# Investigating the role of undercoordinated Pt sites at the surface of layered PtTe$_2$ for methanol decomposition

Jing-Wen Hsueh[1], Lai-Hsiang Kuo[1], Po-Han Chen[2], Wan-Hsin Chen[3], Chi-Yao Chuang[3], Chia-Nung Kuo[4,5], Chin-Shan Lue[4,5,6], Yu-Ling Lai[7], Bo-Hong Liu[7], Chia-Hsin Wang [7], Yao-Jane Hsu [7], Chun-Liang Lin [3] ✉, Jyh-Pin Chou [8] ✉ & Meng-Fan Luo[1] ✉

Transition metal dichalcogenides, by virtue of their two-dimensional structures, could provide the largest active surface for reactions with minimal materials consumed, which has long been pursued in the design of ideal catalysts. Nevertheless, their structurally perfect basal planes are typically inert; their surface defects, such as under-coordinated atoms at the surfaces or edges, can instead serve as catalytically active centers. Here we show a reaction probability > 90 % for adsorbed methanol (CH$_3$OH) on under-coordinated Pt sites at surface Te vacancies, produced with Ar$^+$ bombardment, on layered PtTe$_2$ − approximately 60 % of the methanol decompose to surface intermediates CH$_x$O (x = 2, 3) and 35 % to CH$_x$ (x = 1, 2), and an ultimate production of gaseous molecular hydrogen, methane, water and formaldehyde. The characteristic reactivity is attributed to both the triangular positioning and varied degrees of oxidation of the under-coordinated Pt at Te vacancies.

Two-dimensional transition metal dichalcogenides (TMDs) have attracted considerable interest, owing to their distinctive electronic properties[1–8] and highly tunable surface reactivity[9–16]. TMDs, by virtue of their two-dimensional structures, could provide the largest active surface for reactions with minimal materials consumed, which has long been a goal in the design of ideal catalysts. Structurally perfect basal planes of TMDs are typically inert whereas their surface defects, such as unsaturated coordinative atoms at the surfaces or edges, become catalytically active centers. Many studies on MoS$_2$, one of the most developed TMDs, have demonstrated that by controlling the density of sulfur vacancies (active centers) at MoS$_2$ surface, its reactivity toward varied reactions can be manipulated[13–21]. For instance, the sulfur vacancies at MoS$_2$ surface,

in spite of varied generation approaches (plasma[10], ion bombardment[13] or chemical etching[14]), facilitated hydrogen evolution reaction, through the mechanism involved with altered surface electronic structures and boosted electric conductivity[10,13,14]. Pt-based catalysts, including single crystals, nanoclusters, alloys, and electrodes, have been extensively investigated in both heterogeneous catalysis and electrocatalysis, because of their superior catalytic properties[22–42]. Nevertheless, the catalytic properties of Pt-based TMDs have been little explored[11,43–45], even though they could possess better chemical and structural stability than commercial Pt-based catalysts such as supported Pt nanoclusters. The present study aims to shed light on the catalytic properties of Pt-based TMDs and their potential as catalysts.

[1]Department of Physics, National Central University, No. 300 Jhongda Rd., Jhongli District, Taoyuan City 320317, Taiwan. [2]Department of Materials Science and Engineering, National Tsing Hua University, 101, Section 2 Kuang-Fu Road, Hsinchu 300044, Taiwan. [3]Department of Electrophysics, National Yang Ming Chiao Tung University, No. 1001 University Rd., Hsinchu 300039, Taiwan. [4]Department of Physics, National Cheng Kung University, No. 1 University Rd., Tainan 701, Taiwan. [5]Taiwan Consortium of Emergent Crystalline Materials, National Science and Technology Council, Taipei 10601, Taiwan. [6]Program on Key Materials, Academy of Innovative Semiconductor and Sustainable Manufacturing, National Cheng Kung University, Tainan 701, Taiwan. [7]National Synchrotron Radiation Research Center, 101 Hsin-Ann Rd., Hsinchu Science Park, Hsinchu 300092, Taiwan. [8]Department of Physics, National Changhua University of Education, No. 1, Jin-De Rd., Changhua 50007, Taiwan. ✉e-mail: clin@nycu.edu.tw; jpchou@cc.ncue.edu.tw; mfl28@phy.ncu.edu.tw

We studied the decomposition of methanol (methanol-d$_4$) on layered PtTe$_2$ with various surface-probe techniques under both ultrahigh-vacuum (UHV) and near-ambient-pressure (NAP) conditions. Methanol (CH$_3$OH) decomposition serves as the principal reaction in direct methanol fuel cells (DMFCs), a promising device that converts methanol efficiently to electricity[46–52], and the liberation of hydrogen from the decomposition leads to viable generation of hydrogen, another clean source of energy[53,54]. PtTe$_2$, as a group-10 TMD material, draws much attention for its tunable bandgap, high charge mobility, and ultrahigh air stability[1,3,5–7,11,43,44], but its surface reactivity is little understood. We generated the surface defects on PtTe$_2$ and controlled their concentration with Ar ion (Ar$^+$) bombardment. The surface structures of PtTe$_2$ were characterized using scanning tunneling microscopy (STM) and synchrotron-based photoelectron spectroscopy (PES), while the reactions, intermediates, and products, were monitored using PES, near-ambient-pressure photoelectron spectroscopy (NAP-PES) and near-ambient-pressure mass spectroscopy (NAP-MS). We also performed calculations based on density functional theory (DFT) to explore the mechanisms in detail.

The results showed that under-coordinated Pt (denoted as Pt$_{uc}$) at the PtTe$_2$ surface served as active sites for the reaction. Both dehydrogenation and C-O bond scission occurred for adsorbed methanol on the Pt$_{uc}$ sites, leading to the formation of CH$_x$O* (x = 2 and 3; * denotes adsorbates) and CH$_x$* (x = 1 and 2) as major intermediates and finally production of gaseous molecular hydrogen, methane, water, and formaldehyde via various processes. Gaseous formaldehyde (CH$_2$O$_{(g)}$) was rarely observed from methanol decomposition on either Pt single crystals or supported Pt nanoclusters; meanwhile, CO* (or CO$_{(g)}$), a common product from Pt-based catalysts[22–26,30,33–36,42], was not produced in the present reaction; the pathway of C-O bond scission in the present reaction accounted for a significant proportion (reflected on a considerable quantity of produced CH$_x$*), which also contrasts with its minor role on supported Pt clusters[34]. The catalytic nature of Pt$_{uc}$ sites on PtTe$_2$ thus differs from that of typical Pt-based catalysts. Moreover, the reactivity depended notably on the Pt$_{uc}$ concentration. At a small Pt$_{uc}$ concentration ($\leq$ 10 %), methanol on the Pt$_{uc}$ at surface Te vacancies, the dominating surface defects, decomposed at a great probability (> 90 %). With increased Pt$_{uc}$ concentration (10 − 20 %), the probability decreased as the probability of decomposition to CH$_x$O* was selectively decreased, attributed to structurally different Pt$_{uc}$ generated by extended Ar$^+$ bombardment. Nevertheless, in either case, the reaction probability ($\geq$ 80 %) exceeds those on Pt single crystals[22,25] and supported Pt clusters[34]. We propose, with the support of DFT modeling, that both structural and electronic effects play essential roles in determining the observed catalytic properties. These results suggest that a PtTe$_2$ surface with the Pt$_{uc}$ at surface Te vacancies can serve as a superior catalyst for methanol decomposition; its catalytic selectivity can be controlled with the surface structures manipulated by varied Ar$^+$ bombardment.

## Results
### Structural characterization with STM and PES
We utilized STM to characterize the surface structures of layered PtTe$_2$ before and after the treatment of Ar$^+$ bombardment. The as-cleaved surface was generally very flat and had few defects, as illustrated in Fig. 1a; the hexagonally arranged white spots were topmost Te atoms imaged, and several types of intrinsic defects observed previously with STM[55], such as Te and Pt vacancies, were also observed in the present sample (Fig. S1). After controlled Ar$^+$ bombardment, the surface Te vacancies were evidently increased, together with few small clusters on the surface; the size of the vacancies varied from single- to multiple-Te vacancies (Fig. 1b) and the proportion of larger vacancies increased with the bombardment time (Fig. S1). Continuing to increase either the bombardment time or the Ar$^+$ kinetic energy generated more not only

the surface Te vacancies but also structural variations, such as island edges and re-deposited atoms (Fig. 1c, d). With the aim of correlating structures with reactivity, we chose a kinetic energy of 0.5 keV for incident Ar$^+$ and a reduced Ar$^+$ dosage (the sample current multiplied by bombardment time) to control structural complexity for reaction experiments − the Te vacancies were produced as the main surface defects while the surface crystallinity (monitored with the RHEED measurements, Fig. S2) was largely sustained.

Fig. 1e–h exemplify a high-resolution image for two single-Te vacancies and the corresponding line profile across one of the vacancies. The vacancy depth about 0.1 nm (Fig. 1f), similar to that of an intrinsic Te vacancy at the surface (Fig. S1), and the well overlap of the top Te atoms and vacancies in the model with the imaged ones (Fig. 1g) suggest that the vacancies were formed by the removal of the topmost Te atoms, i.e., the surface Te vacancies. As the positions of the surface Te atoms neighboring the vacancies remained nearly unchanged (Fig. 1g) and as they were mainly bonded to the underlying Pt, the generation of these Te vacancies altered little the positions of underlying Pt atoms. The well match of the STM image with the DFT-simulated one (Fig. 1h), produced based on the vacancy model in Fig. 1g, corroborates the vacancy structure. Small Ar$^+$ dosages removed mainly the surface Te while left the underlying Pt atoms unaltered. Our DFT modeling for the reactions discussed below also adds weight to the argument. The small clusters in Fig. 1b likely correspond to re-deposited atoms (0.15 nm, one-atom high) after Ar$^+$ bombardment. As they were much fewer than the surface Te vacancies at small Ar$^+$ dosages, for which the reaction experiments were primarily performed, their contribution to the observed reactions was considered limited even though they could initiate or assist (as adsorption sites) reactions.

Our PES spectra further characterized these surface defects observed with STM. Fig. 2a exemplifies the comparison of the PES spectra of Pt 4 f core level for the layered PtTe$_2$ as cleaved and bombarded by Ar$^+$. The spectrum from as-cleaved PtTe$_2$ (upper panel of Fig. 2a) shows a doublet, Pt 4f$_{5/2}$ and Pt 4f$_{7/2}$, centered at the binding energies (BE) 75.9 and 72.6 eV respectively, which corresponds to intact Pt-Te chemical bonds in PtTe$_2$. Meanwhile, the PES spectrum from Ar$^+$-bombarded PtTe$_2$ shows an additional feature at a BE slightly smaller than that of the main Pt feature, i.e., at 71.8 eV for Pt 4f$_{7/2}$ and 75.1 eV for Pt 4 f$_{5/2}$ (indicated by red fitted curves in the bottom panel). The smaller BE for Pt 4 f implies less oxidized Pt, corresponding to under-coordinated Pt (Pt$_{uc}$), at the PtTe$_2$ surface and therefore the removal of Te atoms by Ar$^+$ bombardment. In contrast, the Te 4d signals were insensitive to the removal of neighboring Te atoms —— the line shape of Te 4d doublet altered little after Ar$^+$ bombardment (Fig. 2b), since the surface Te is mainly bonded to the underlying Pt. Both the Pt 4f and Te 4d lines from Ar$^+$-bombarded PtTe$_2$ shifted negatively (approximately 0.1 eV as shown in Fig. 2a, b), likely because the bombardment induced a band-bending effect. As the spectral features for varied Ar$^+$ dosages are similar despite varied intensities of the Pt$_{uc}$ features (Fig. S3), the Pt$_{uc}$ signals correspond to the Pt$_{uc}$ at surface Te vacancies (denoted as Pt$_{uc}$-Vac) and also to those at other surface defects (denoted as Pt$_{uc}$-Ex) generated by greater Ar$^+$ dosages. Our experiments show that the number or concentration of Pt$_{uc}$ can be controlled with Ar$^+$ dosage and Ar$^+$ kinetic energy. Fig. 2c plots the ratio of the Pt$_{uc}$ and total Pt signals (denoted as Pt$_{uc}$/Pt), measured by the integrated intensities of the red and black fitted curves (Fig. 2a), as a function of the Ar$^+$ dosage. The Pt$_{uc}$/Pt ratio increased almost in a linear fashion with the Ar$^+$ dosage, despite varied Ar$^+$ kinetic energies. As about 90 % of the Pt signals came from the top two PtTe$_2$ bilayers (according to the escape depth of the Pt 4 f photoelectrons) and as Ar$^+$ bombardment at a small dosage removed primarily the surface Te (Fig. 1), each Pt$_{uc}$/Pt ratio corresponds to a derivable concentration of surface Pt$_{uc}$. Notably, the rate of increase of the Pt$_{uc}$/Pt ratio depended on the Ar$^+$ kinetic energy, since the cross-section of removing surface

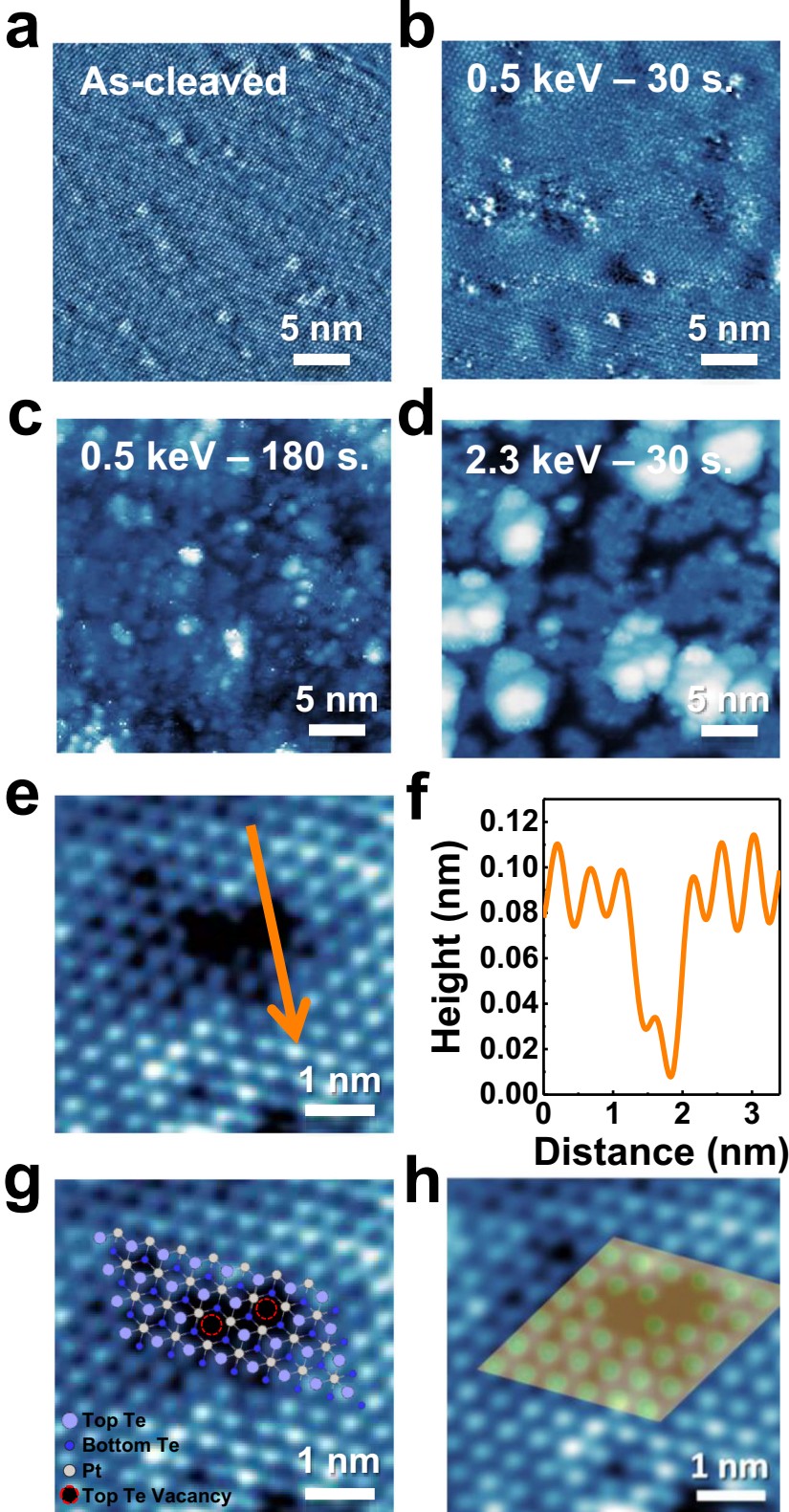

**Fig. 1 | Charcterization of surface structures of layered PtTe$_2$ with STM.** STM images for layered PtTe$_2$ **a** as-cleaved (V$_s$ = -250 mV, I$_t$ = 1.35 nA) and bombarded by Ar$^+$ with **b** 0.5 keV for 30 sec. (V$_s$ = −150 mV, I$_t$ = 1.20 nA) and **c** 180 sec., and with **d** 2.3 keV for 30 sec. (V$_s$ = −500 mV, I$_t$ = 2.90 nA); **e** the high-resolution image of two single-Te vacancies, **f** the line profile across a single-Te vacancy, **g** the overlap of the two single-Te vacancies model with the imaged ones and **h** the match of the STM image with the DFT-simulated one produced based on the vacancy model in (**g**). In **g**, light blue, blue and grey balls denote top, bottom Te and Pt atoms in the topmost PtTe$_2$ bilayer; the dash-line circles denote single-Te vacancies. In **h**, green balls denote the simulated images for the top Te atoms; the simulated image was derived with bias −0.1 V while the images obtained with −0.5 ‑ −0.1 V were similar.

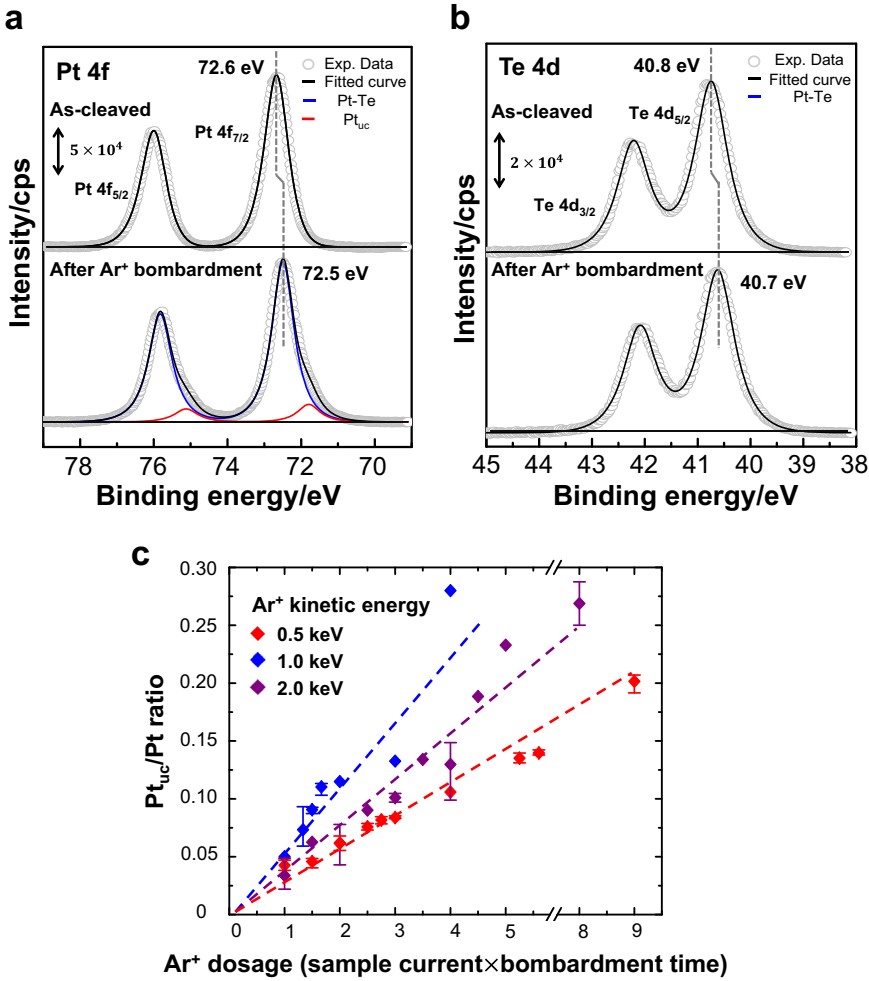

**Fig. 2 | Charcterization of surface structures of layered PtTe₂ with PES.** PES spectra of **a** Pt 4 f and **b** Te 4d core levels from layered PtTe₂ as cleaved and bombarded by Ar⁺ (0.5 keV, 3 mins); **c** ratios of integrated intensities of $Pt_{uc}$ to Pt 4 f lines as a function of Ar⁺ dosages. In **a** and **b**, gray circles denote the spectra and black lines the sum of fitted curves; the signals from intact Pt and under-coordinated Pt ($Pt_{uc}$) in the layered PtTe₂ are fitted with blue and red lines, respectively. In **c**, the Ar⁺ dosages were calculated by sample current multiplied by sputtering time; the data obtained with Ar⁺ kinetic energies of 0.5, 1.0, and 2.0 keV were represented by red, blue, and purple rhombuses respectively in the plot, and the error bars indicate the reproducibility.

Te varied with the Ar⁺ kinetic energy. The Ar⁺ at 0.5 keV was chosen to prepare the sample, as it exhibited the best controllability in producing small $Pt_{uc}$ concentrations (small $Pt_{uc}$/Pt ratios), warranting the surface Te vacancies as the dominating surface defects for catalytic studies (Fig. S1). Our reaction experiments were primarily performed on the PtTe₂ with $Pt_{uc}$/Pt ratios ≤ 0.13 (Ar⁺ dosage ≤ 4.5, Fig. 2c), corresponding to the $Pt_{uc}$ concentration ≤ 20 %; the Ar⁺ dosages (0.5 keV) at 1 and 6 in Fig. 2c produced the surfaces resembling those shown in Fig. 1b, c respectively.

**Methanol decomposition monitored with PES, NAP-PES and NAP-MS**
The methanol reactions were characterized primarily by PES spectra; with the spectra, we monitored the evolution of surface species with temperature and $Pt_{uc}$ concentrations. Fig. 3a, b compare the C 1s spectra for methanol adsorbed on as-cleaved and Ar⁺-bombarded PtTe₂ ($Pt_{uc}$/Pt ratio = 0.07) at 145 K and annealed stepwise to selected temperatures. The dominant feature at 145 K on either surface, centered about 286.5 eV, is assigned to the C 1s line of monolayer methanol adsorbed on PtTe₂ (top in Fig. 3a, b), since multilayer methanol desorbed near 130 K[34,56,57]. On the as-cleaved PtTe₂ surface, the methanol signals decreased with increased temperature and vanished at 200 K, reflecting the desorption of methanol (Fig. 3a). In contrast, on the PtTe₂ with a number of surface $Pt_{uc}$ produced by Ar⁺

bombardment, new features grew at 283.8 – 285.1 eV above 160 K, at the expense of attenuating methanol feature at 286.5 eV (Fig. 3b). The results suggest that with increased temperature, a fraction of methanol desorbed whereas the other fraction decomposed and produced new carbon species. As the as-cleaved PtTe₂ has very limited surface defects, the contrasting results indicate that the structurally perfect basal plane of layered PtTe₂ is catalytically inert but the surface $Pt_{uc}$, mostly at the Te vacancies (Fig. 1b, e), i.e. $Pt_{uc}$-Vac, are reactive toward methanol decomposition. The products at elevated temperature consisted of three carbon species: $CH_xO^*$ (x = 1-3), $CH_x^*$ (x = 1-3) and atomic carbon ($C^*$), corresponding to the C 1s lines centered at 285.1[58-64], 283.8[65-67] and 284.2 eV (Fig. S4a), respectively. CO* was not expected because our adsorption experiments showed no CO adsorbed on such a defective PtTe₂ surface even at 145 K. Fig. 3c, d exemplify the fits to the C 1s lines (at 180 and 260 K) with characteristic fitted curves representing C 1s signals from adsorbed monolayer methanol and the proposed products. $CH_xO^*$ (red curve in Fig. 3c, d) and $CH_x^*$ (blue) were the primary products at 160 – 260 K but $C^*$ became notable above 260 K, implying that further decomposition of $CH_xO^*$ and/or $CH_x^*$ occurred at elevated temperature. Fig. 3e plots the integrated intensities of these fitted C 1s curves, used to measure the quantities of produced $CH_xO^*$, $CH_x^*$, and $C^*$, as a function of temperature. The produced $CH_xO^*$ (red circles) and $CH_x^*$ (blue) increased to maxima near 180 K and decreased at even higher temperatures, whereas $C^*$

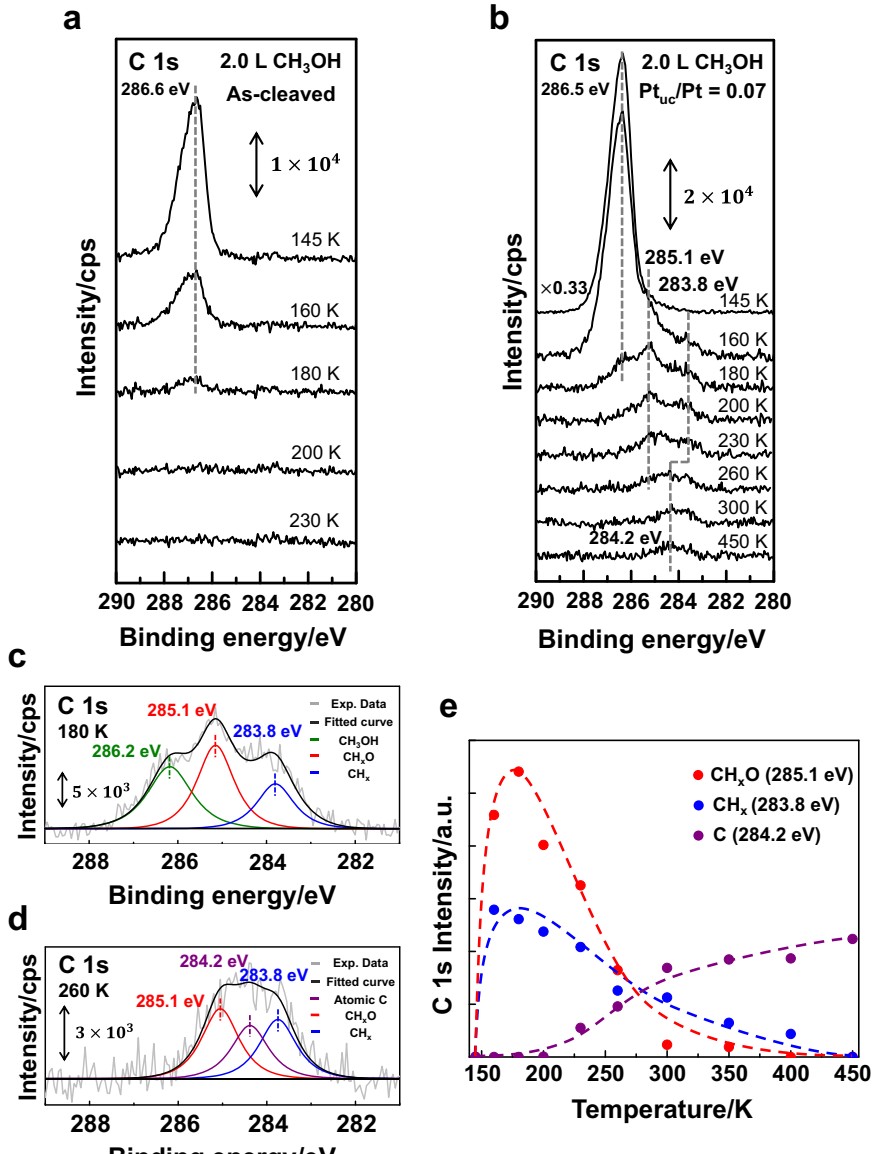

**Fig. 3 | Reactions of methanol adsorbed on PtTe₂ monitored with PES.** PES spectra of C 1 s core level for methanol (4.0 and 2.0 L) adsorbed on **a** as-cleaved and **b** Ar⁺-bombarded PtTe₂ at 145 K and annealed stepwise to selected temperatures; the illustration fits the spectra obtained at **c** 180 and **d** 260 K; **e** the integrated intensities of the fitted curves representing CHₓO* (red circles), CHₓ* (blue) and C*

(purple) signals as a function of temperature. In **c** and **d**, gray lines denote the spectra and black ones the sum of fitted curves; absorbed methanol, CHₓO*, CHₓ*, and C* signals are fitted with green, red, blue, and purple lines, respectively. The dash lines in (**e**) are the guidance for eyes only.

(purple) began to emerge above 200 K and became the major species at and above 300 K. The formation of CHₓO* and CHₓ* indicates that dehydrogenation and C-O bond scission, the two competing processes of methanol decomposition[30,34,35,68], were both catalytically activated at such low temperature on the surface Ptᵤ꜀ sites. A fraction of CHₓO* and CHₓ* must have desorbed at elevated temperature, since the quantity of the remaining C* was not comparable to that of the produced CHₓO* and CHₓ* (Fig. 3e). Our NAP experiments presented below indicate that the desorbing carbon species consisted largely of CH₂O(g) and CH₄(g).

To reveal how the catalytic properties of PtTe₂ surface vary with the concentration of surface Ptᵤ꜀, we plot the probabilities of conversion of adsorbed monolayer methanol to CHₓO* (red circles) and CHₓ* (blue) as a function of the concentration of surface Ptᵤ꜀, shown in Fig. 4a. The conversion probability was derived from the ratio of the C 1 s intensities of CHₓO* (or CHₓ*) to monolayer methanol (145 K); the former was measured at 180 K because the maximum quantity of CHₓO* (or CHₓ*) was produced around 180 K; only monolayer

methanol was considered because it was directly in contact with the PtTe₂ surface. The concentration of surface Ptᵤ꜀ was estimated according to the measured Ptᵤ꜀/Pt ratios (Fig. 2), as mentioned above; the percentage corresponds to the fraction of Ptᵤ꜀ in the total amount of Pt in the top PtTe₂ bilayer. For CHₓ* (blue circles), the conversion probability of monolayer methanol increased almost in linear proportion to the Ptᵤ꜀ concentration, corroborating that Ptᵤ꜀ served as reactive sites for the C-O bond scission. Nevertheless, the trend becomes complicated for CHₓO* (red). The conversion probability increased linearly at a Ptᵤ꜀ concentration ≤ 10 %, while became saturated (10 - 20 %) or even decreased at a greater Ptᵤ꜀ concentration (> 20 %). At greater Ptᵤ꜀ concentrations (> 10 %) the Ptᵤ꜀ structurally different from Ptᵤ꜀-Vac had grown and accounted for a fraction of total Ptᵤ꜀, although they were not resolved in the Pt 4 f spectra. Such Ptᵤ꜀ corresponded to Ptᵤ꜀-Ex; they possessed different catalytic properties so initiated a separate reaction pathway, for which the C-O bond scission was more facilitated so CHₓO* became instable and decreased.

**a**

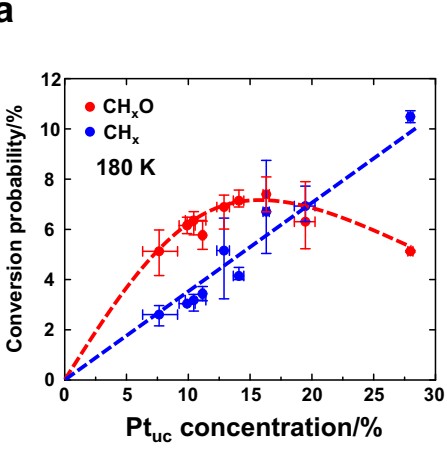

**b**

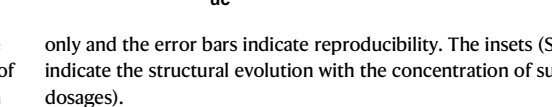

**Fig. 4 | Varied reaction probabilities of methanol adsorbed on PtTe$_2$ surface with Pt$_{uc}$ sites.** Probabilities of conversion to CH$_x$O* (red circles) and CH$_x$* (blue) of monolayer methanol adsorbed on **a** PtTe$_2$ surface, and those on **b** Pt$_{uc}$ sites as a function of the concentration of surface Pt$_{uc}$. The dash lines are drawn to guide eyes only and the error bars indicate reproducibility. The insets (STM images) in (**b**) indicate the structural evolution with the concentration of surface Pt$_{uc}$ (Ar$^+$ dosages).

The conversion probability of methanol adsorbed on the Pt$_{uc}$ sites (denoted as methanol/Pt$_{uc}$) as a function of the Pt$_{uc}$ concentration shows clearly the evolving catalytic selectivity (Fig. 4b). The fraction of methanol/Pt$_{uc}$ in total monolayer methanol was estimated to be that of Pt$_{uc}$ in total amount of Pt in the top PtTe$_2$ bilayer (the Pt$_{uc}$ concentration), by assuming that monolayer methanol adsorbed uniformly on the PtTe$_2$ surface. As shown in Fig. 4b, the conversion probability of methanol/Pt$_{uc}$ to CH$_x$* varied little with the Pt$_{uc}$ concentration, remaining near 35 %; in contrast, that to CH$_x$O* was approximately 60 % at a Pt$_{uc}$ concentration ≤ 10 % but decreased continuously at a Pt$_{uc}$ concentration > 10 %. The observation does not simply suggest that the formation of CH$_x$O* (dehydrogenation) was structure-sensitive whereas that of CH$_x$* (C-O bond scission) was not[69–71] because the productions of these two intermediates were not necessarily separate − CH$_x$* could be produced largely via the C-O bond scission of CH$_x$O*. Besides, both the dehydrogenation and C-O bond scission were considered sensitive to structures[69–71]. The present result is likely due to enhanced C-O bond scission, which decreases CH$_x$O* but increases CH$_x$*, and obstructed dehydrogenation, which decreases both CH$_x$O* and CH$_x$*, on the Pt$_{uc}$-Ex sites mentioned above. Our STM measurements show that at a greater Ar$^+$ dosage (Pt$_{uc}$ concentration > 10 %), the generated surface defects include not only the Te vacancies (Fig. 1b) but also other defects, such as edges of PtTe$_2$ patches (islands) and Pt-Te nanoclusters formed by nucleation of redeposited Pt and Te (exemplified in Fig. 1c and Fig. S4b). We thus associated the Pt$_{uc}$-Ex with the Pt$_{uc}$ at these two surface defects. Nevertheless, we note that the total reaction probability was remarkably great − about 95 % at a Pt$_{uc}$ concentration ≤ 10 % and remaining greater than 80 % even at a concentration near 20 %. In either case, the reaction probability exceeded that for methanol on either Pt single-crystal surfaces (10 %)[22,25] or supported Pt clusters (60 - 70 %)[34]. Additionally, the probability of conversion to CH$_x$* (35 %) was also evidently greater than that on Pt clusters (<10 %)[34]; the C-O bond scission pathway apparently played an important role in the present reaction.

The above experiments demonstrate that the catalytic reactivity of layered PtTe$_2$ is controllable by the surface Pt$_{uc}$ under UHV conditions. To unveil their catalytic performance under "real-world" conditions[72], we investigated methanol reactions on layered PtTe$_2$ near ambient pressure. Fig. 5a exemplifies the NAP-PES spectra of C 1 s core level from PtTe$_2$ bombarded by Ar$^+$ (Pt$_{uc}$/Pt ratio = 0.10) and subsequently exposed stepwise to selected pressures of methanol at 300 K; 300 K was used because the desorption and further decomposition of

products (intermediates) already occurred (Fig. 3). The as-bombarded PtTe$_2$ surface was free of carbon contamination, indicated by the absence of C 1 s signals (the bottom of Fig. 5a); increasing methanol pressure to 10$^{-4}$ mbar, a small C 1 s line arose around 284.2 eV and continued to grow with increased pressure; at 10$^{-3}$ mbar and above, a shoulder centered about 283.0 eV (Fig. 5a,b) also grew. The former is assigned to C*, while the latter to CH$_x$*, both of which resulted from decomposed methanol on PtTe$_2$. The reactions must have occurred on the surface Pt$_{uc}$ sites (largely Pt$_{uc}$-Vac sites at this small Pt$_{uc}$ concentration) as the experiments on as-cleaved PtTe$_2$ (with scarce surface Pt$_{uc}$), as a comparison, showed negligible C 1 s signals. This observation agrees with that on the Ar$^+$-bombarded PtTe$_2$ surface under UHV conditions, in which C* and CH$_x$* were primary remaining species on the surface at 300 K (Fig. 3e). These C 1 s signals increased with methanol pressure, since more methanol decomposed on the PtTe$_2$ surface at greater methanol pressures. They decreased at 0.1 mbar (second from the top in Fig. 5a) as the photoelectrons were attenuated by the increased pressure; at such a great pressure, the C 1 s feature resulting from gaseous methanol also appeared at 288.5 eV[35,73]. Notably, the ratio of CH$_x$* to C* signals increased with methanol pressure but altered little with the decreasing pressure from 0.1 to 10$^{-7}$ mbar. The ratio was affected little by the pressure-induced signal attenuation but determined by the composition of carbon species on the surface. The fraction of CH$_x$* in total surface carbon species became greater at a greater pressure or C* concentration. We speculate that a greater C* concentration at the Pt$_{uc}$ sites suppressed the dehydrogenation of CH$_x$* by altering the electronic properties of Pt$_{uc}$ and/or adsorption configurations of CH$_x$*.

The corresponding gaseous products from the methanol reactions near ambient pressure were monitored with NAP-MS. Fig. 5c exemplifies the observed main products, including D$_{2(g)}$, D$_2$O$_{(g)}$/CD$_{4(g)}$ and CD$_2$O$_{(g)}$, from Ar$^+$-bombarded PtTe$_2$ (Pt$_{uc}$/Pt ratio = 0.10) at 300 K exposed to methanol-d$_4$ (CD$_3$OD) at varied pressures. Instead of methanol, methanol-d$_4$ was used because these isotopic variants have similar chemical properties, such as the activation for desorption and decomposition (determined from their electronic structures), but methanol-d$_4$ gave clearer signals of molecular deuterium (D$_2$), an essential product to reveal the reaction mechanisms, in desorption experiments. These three desorbing species increased generally with methanol-d$_4$ pressure, consistent with the above NAP-PES experiments, as more methanol-d$_4$ interacted with the Pt$_{uc}$ at a higher methanol-d$_4$ pressure. D$_{2(g)}$ came from recombinative desorption of D*, while D* was produced from dehydrogenated CD$_3$OD*,

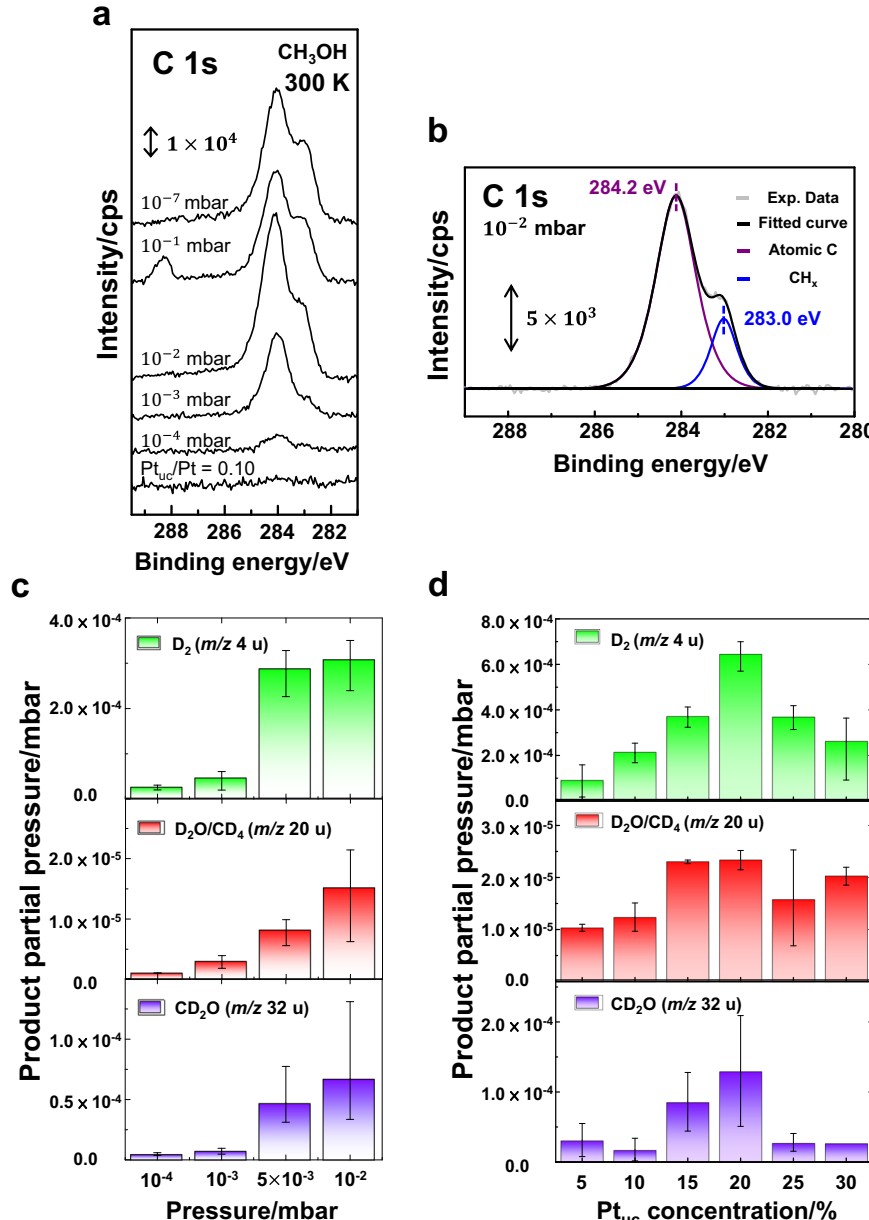

**Fig. 5 | Reactions of methanol on PtTe$_2$ under NAP conditions monitored with NAP-PES and NAP-MS.** NAP-PES spectra of **a** C 1s core level from Ar$^+$-bombarded PtTe$_2$ (Pt$_{uc}$/Pt ratio 0.10) exposed to varied pressures of methanol, as indicated, at 300 K; **b** the illustration of the fit to the spectrum obtained at methanol pressure 10$^{-2}$ mbar; the production of D$_{2(g)}$ (*m/z* 4 u), D$_2$O$_{(g)}$/CD$_{4(g)}$ (*m/z* 20 u) and CD$_2$O$_{(g)}$ (*m/z* 32 u) from PtTe$_2$ at 300 K, as a function of **c** methanol-d$_4$ pressure (Pt$_{uc}$/Pt ratio 0.10) and **d** Pt$_{uc}$ concentration (10$^{-2}$ mbar). In (b), the gray line denotes the spectrum, and the black one is the sum of fitted curves; CH$_x$* and C* signals are fitted with blue and purple lines, respectively. In **c** and **d**, the error bars indicate the reproducibility.

CD$_x$O*, and CD$_x$*; therefore, its formation as the major desorbing species reflects the essential role of dehydrogenation at different stages of methanol-d$_4$ decomposition. Both D$_2$O$_{(g)}$ and CD$_{4(g)}$ were possible products, reflecting the process of C-O bond scission, but they could not be resolved by our mass spectra. CD$_{4(g)}$ was formed through CD$_x$* combining with D*, and D$_2$O$_{(g)}$ through either O* or OD* combining with D*. The observed CD$_2$O$_{(g)}$ suggests that CH$_x$O* decreased above 180 K (Fig. 3e) through not only C-O bond scission but also desorption as CH$_2$O$_{(g)}$. The formation of CD$_2$O$_{(g)}$ and D$_2$O$_{(g)}$ also rationalizes scarce O* remaining on the surface, as evidenced by vanishing O 1s signals above 200 K in the UHV PES experiments as well as absent O 1s signals in the NAP-PES spectra (300 K). The absence of CO$_{(g)}$ and CO* indicates that the dehydrogenation of methanol-d$_4$ (or methanol) to CO, a process typically observed on Pt clusters or single crystals[22–24,26,30,33–36,42], did not occur on the Pt$_{uc}$

sites of PtTe$_2$. The observed C* accordingly originated from dehydrogenated CH$_x$*.

The gaseous products also confirm that the reactivity and selectivity of Pt$_{uc}$ sites depend on the Pt$_{uc}$ concentration. Fig. 5d shows the gaseous products (at methanol pressure 10$^{-2}$ mbar) as a function of the Pt$_{uc}$ concentration. For a Pt$_{uc}$ concentration <20 %, the products increased generally with the Pt$_{uc}$ concentration, which is consistent with the trend shown in Fig. 4a. The production of CD$_2$O$_{(g)}$ dropped dramatically when the Pt$_{uc}$ concentration increased above 20 %, in agreement with the decrease of CH$_x$O* (Fig. 4a). The produced D$_{2(g)}$ also decreased, indicating the selectively suppressed dehydrogenation to CH$_x$O*. In contrast, CD$_{4(g)}$/D$_2$O$_{(g)}$ decreased only slightly at a Pt$_{uc}$ concentration > 20 %, as CD$_x$* increased at 180 K (Fig. 4a); the proportion of CD$_x$* desorbing as CD$_{4(g)}$ could vary with the Pt$_{uc}$ concentration. The dependence is associated, as discussed above, with the

composition of $Pt_{uc}$-Vac and $Pt_{uc}$-Ex evolving with the $Pt_{uc}$ concentration.

## DFT modeling

Our first-principles DFT modeling aimed to elucidate the key mechanisms behind the observed reactions. We established a Te-divacancy model by removing two adjacent Te atoms at the topmost layer in order to mimic a $PtTe_2$ surface at which the reactive sites consist primarily of the Te vacancies (Fig. 1b,e). The model is applicable to the cases of $Pt_{uc}$ concentration ≤ 20 % because the Te vacancies remained as the main surface defects despite the growth of the $Pt_{uc}$-Ex at a $Pt_{uc}$ concentration > 10 %. Although the Te vacancies of various sizes were observed, the divacancy model suffices to represent the key features of the reactions on the $Pt_{uc}$-Vac sites. A divacancy site has five $Pt_{uc}$-Vac: one loses two Te-Pt bonds in the bilayer structure (denoted as $Pt_{uc2}$-Vac), corresponding to the coordination to four Te, and the other four lose one Te-Pt bond ($Pt_{uc1}$-Vac), illustrated in Fig. 6a. The structural modeling confirms that the underlying Pt atoms remain nearly at the same positions after the removal of surface Te. We considered three main reaction processes in the methanol decomposition: dehydrogenation from either oxygen (red arrows) or carbon (green arrow), and C-O bond scission (deoxygenation and dihydroxylation; black arrows), presented in Fig. 6b. The adsorption configurations and energies of methanol and its decomposition fragments are presented in Fig. S5. The modeling shows that a methanol molecule adsorbs on a $Pt_{uc}$-Vac site with an O-Pt binding configuration and adsorption energy −1.02 eV, which is evidently stronger than that (−0.41 eV) on pristine $PtTe_2$ basal plane (Fig. S5). The $Pt_{uc}$-Vac, therefore, has the potential to serve as a reactive center on $PtTe_2$. The three main decomposition processes at varied stages at the divacancy sites were calculated and compared. For the first step of decomposition, the barrier for dehydrogenation from oxygen (0.66 eV) of $CH_3OH^*$ was significantly smaller than those for dehydrogenation from carbon (1.42 eV) and dehydroxylation (1.82 eV). The comparison suggests the preferential formation of methoxy ($CH_3O^*$), agreeing well with the observed selectivity that methanol decomposed via a pathway to produce more $CH_xO^*$ than $CH_x^*$ at a small $Pt_{uc}$ concentration (Fig. 4a). The great difference between the energy barriers for desorption and dehydrogenation to $CH_3O^*$ (1.02 vs. 0.66 eV) also explains the great conversion probability for methanol adsorbed on the $Pt_{uc}$-Vac site (Fig. 4b). $CH_3O^*$ is expected to undergo further dehydrogenation (with a barrier 1.08 eV) to formaldehyde ($CH_2O^*$), because of the considerably greater activation energies for the two competing processes, namely desorption (2.58 eV) and deoxygenation (2.29 eV). $CH_2O^*$, due to its planar structure with $sp^2$ hybridized carbons, has an adsorption energy (−1.52 eV) smaller than those of the other intermediates, that impedes subsequent decomposition. Compared to the greater barriers for its C-O bond cleavage (3.33 eV) or further dehydrogenation (to $CHO^*$, 1.56 eV), $CH_2O^*$ would prefer desorption (as $CH_2O_{(g)}$) to decomposition at elevated temperature, as observed in our NAP-MS experiments (Fig. 5c, d). Alternatively, if $H^*$ from the dehydrogenation is nearby (not yet desorbed as $H_{2(g)}$), then $CH_2O^*$ could also combine with $H^*$ to form $CH_2OH^*$ (with a small barrier 0.12 eV), which provides a feasible pathway for further reactions. $CH_2OH^*$ would decompose via dehydroxylation to $CH_2^*$ and $OH^*$, instead of dehydrogenation to $CHOH^*$ and $H^*$; the latter process does not occur because the inverse process ($CHOH^* + H^* \rightarrow CH_2OH^*$) has a negligible barrier (<0.01 eV) and $CH_2OH^*$ has a lower total energy. As the barrier for the dehydroxylation (1.27 eV) of $CH_2OH^*$ is evidently smaller than those for the C-O bond scission of $CH_3O^*$ and $CH_2O^*$ (2.29 and 3.33 eV), the observed $CH_x$ species resulted from decomposed $CH_2OH^*$. As a result, the observed $CH_xO^*$ species in our PES spectra correspond to $CH_3O^*$ and $CH_2O^*$ and the $CH_x$ species mainly to $CH_2^*$ and $CH_1^*$. The $CH_2^*$ species may undergo either further dehydrogenation to yield $C^*$ or combination with $H^*$ to produce $CH_{4(g)}$, as observed in NAP-PES spectra (Fig. 5a)

and NAP-MS spectra (Fig. 5c, d) respectively. Details of the calculated energy barriers are provided in our Supplementary Information (Figs. S5 – S18).

After the C-O bond scission of $CH_2OH^*$, the produced $OH^*$ either diffuses away from the active sites or combines with $H^*$ to desorb as $H_2O_{(g)}$ (Fig. 5c, d), so its "poisoning effect" is insignificant. The calculated activation energy for $OH^*$ to migrate to an intermediate adsorption site on Te at the edge of divacancy amounts to 0.95 eV; their diffusion barrier on the basal plane is even smaller (0.36 eV), as shown in Fig. S19. These diffusion barriers are smaller than that for the C-O bond scission of $CH_2OH^*$, so with the progress of methanol decomposition, the produced $OH^*$ can diffuse readily to other sites to prevent the active sites from obstruction.

The above modeling shows that $Pt_{uc}$-Vac in the divacancy model activates the decomposition in a coordinative manner; the adsorbates ($CH_3OH^*$, its decomposition intermediates and fragments) are bonded mostly to two or three $Pt_{uc}$-Vac in varied decomposition processes (Figs. S5 – S18). Fig. 7a shows the simulated dehydrogenation of $CH_3OH^*$ to $CH_3O^*$ as an example. $CH_3OH^*$ first adsorbs on the $Pt_{uc2}$-Vac through its O and then undergoes the scission of O-H bond. At the final stage, $H^*$ is bonded to one of $Pt_{uc1}$-Vac and $CH_3O^*$ to both $Pt_{uc1}$-Vac and $Pt_{uc2}$-Vac. It is noted that neither $CH_3O^*$ nor $H^*$ is bonded to Te near the vacancy in the process, since they adsorb weakly on these Te sites, like $CH_3OH^*$ and its other decomposition intermediates or fragments (Fig. S5). As Te atoms at $PtTe_2$ surface are mainly bonded to the underlying Pt[6,11,55], instead of their neighboring Te atoms, the removal of surface Te atoms altered little the electronic properties of Te atoms surrounding the vacancies —— the bonding of these surface Te atoms remains saturated. As a result, they were not heavily involved in the methanol decomposition. In the divacancy model, $Pt_{uc2}$-Vac appears to be more active than $Pt_{uc1}$-Vac, indicating that the processes are mostly centered around $Pt_{uc2}$-Vac in spite of varied reaction pathways (Figs. S5 –S18). The $Pt_{uc}$-Vac are like separated Pt single atoms, triangularly positioned and oxidized to different extents dependent on their bonding to Te. The number of coordination (to Te) or missing Te-Pt bonds of $Pt_{uc}$-Vac determines its electronic properties and hence catalytic properties. To illuminate the effect of the coordination number of $Pt_{uc}$-Vac on the activity, a tri-vacancy model with a $Pt_{uc3}$-Vac in the middle of the tri-vacancy site, which was also likely formed on the $Ar^+$-bombarded $PtTe_2$ surface, was introduced to simulate the reactions. The results show that the adsorption energies of $CH_3OH^*$, $CH_3O^*$ and $CH_2O^*$ on the tri-vacancy site were slightly increased by 0.05 - 0.10 eV, and the energy barrier for dehydrogenation of $CH_3OH^*$ to $CH_3O^*$ remained similar, whereas that of $CH_3O^*$ to $CH_2O^*$ was reduced by 0.12 eV (Figs. S20−S22). Consequently, the formation of $CH_2O^*$ on this tri-vacancy became more probable. Reducing the coordination to Te enhances the activity of $Pt_{uc}$-Vac.

The corresponding electronic structures reflect the same trend. The measured and calculated local densities of states (LDOS) near the Fermi level of $Pt_{uc0-3}$-Vac (Fig. 7b) show consistently that with decreased coordination number, the $Pt_{uc0-3}$-Vac at $PtTe_2$ surface become more metallic, reflected on the enhanced LDOS near the Fermi level; meanwhile, their d-band centers shift toward higher energies (Fig. 7c). Both results are indicative of enhanced catalytic reactivity[74,75]. The LDOS of $Pt_{uc1-3}$-Vac at $PtTe_2$ surface and Pt at Pt(111) surface are evidently different (Fig. S23), accounting for their different catalytic behaviors. Previous studies indicate that the spatial distribution and orientation of frontier orbitals of single-atom catalysts are well correlated with adsorption and catalytic activities[76]. We note that the spatial distributions of frontier orbitals (in the energy ranging from -0.25 eV to the Fermi level) of $Pt_{uc}$-Vac at $PtTe_2$ surface expanded with decreased coordination number, as plotted in the insets of Fig. 7c. The expansion reflects a promoted probability of wave-function overlap, which is required for adsorption and catalytic activities[76], so agrees with the enhanced reactivity indicated above. Thus, the characteristic reactivity

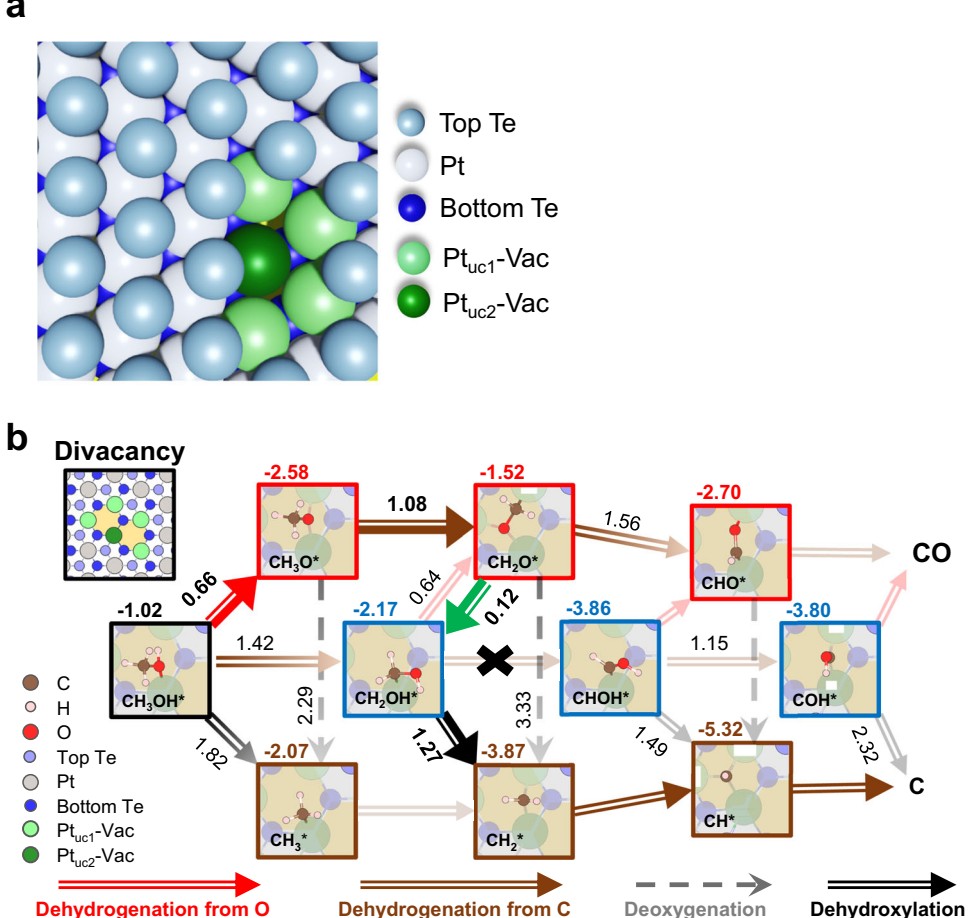

**Fig. 6 | Schematic pathways of methanol reactions on a Te-divacancy site at PtTe₂ surface. a** The atomic model for a Te divacancy at PtTe₂ surface. **b** Pathways of methanol decomposition on the Te-divacancy site at PtTe₂ surface. In **b**, the possible intermediates in their most stable configurations and their adsorption energies on the Te-divacancy site are presented (value at the upper-left corner of each panel). The sizes of Pt and Te atoms in the model are varied to illustrate their relative positions from top view. Brown and red hollow arrows indicate the dehydrogenation process from C and O atoms of methanol, respectively; black hollow and grey dashed arrows represent dehydroxylation and deoxygenation processes, respectively; the green hollow arrow indicates a hydrogenation process. For clarity, the arrows for the most plausible processes in the methanol decomposition are enlarged. The value beside each arrow is provided for the energy barrier of the specific process.

of the $Pt_{uc}$-Vac arises from not only the peculiar structural (geometric) effect − triangularly positioned $Pt_{uc}$-Vac, but also the electronic effect − differently oxidized $Pt_{uc}$-Vac.

The above modeling also indicates that enlarging the Te vacancies (increasing the number of more active $Pt_{uc3}$-Vac) promotes the main reaction processes shown in Fig. 6b, instead of altering the reaction pathway. It accordingly supports the argument that a separated reaction pathway reflected on either the decreased conversion probability to $CH_xO$ on $Pt_{uc}$ sites (Fig. 4b) or the decreased production of $CD_2O_{(g)}$ and $D_{2(g)}$ (Fig. 5d) at a greater $Pt_{uc}$ concentration was initiated by the $Pt_{uc}$-Ex. The $Pt_{uc}$-Ex consists largely of $Pt_{uc}$ at the edges of PtTe₂ patches (Fig. 1c and S4b) or in redeposited Pt-Te nanoclusters (Fig. 1c). The $Pt_{uc}$-Ex at the edges of PtTe₂ patches (islands) are primarily the $Pt_{uc}$ with one missing Te-Pt bond ($Pt_{uc1}$-Ex), as illustrated in Fig. S24. Our DFT modeling for methanol on the edges of PtTe₂ patches (Figs. S24–S27) show that $CH_3OH$* on the edge $Pt_{uc1}$-Ex sites prefers desorption to decomposition, in the light of evidently greater activation energies for the dehydrogenation to $CH_3O$* (1.64 eV) and the C-O bond scission (2.07 eV) than that for desorption (0.57 eV). Notably, the activation energies are greater and the adsorption is weaker than those (Fig. 6b and Fig. S20) on the $Pt_{uc2}$-Vac (or $Pt_{uc3}$-Vac) in the divacancies (trivacancies). The result implies that increasing the edge $Pt_{uc1}$-Ex sites decreases the average conversion probability (to both $CH_xO$* and

$CH_x$*) on $Pt_{uc}$ sites, which explains the observed behavior of $CH_xO$* (Fig. 4b). As a result, the $Pt_{uc}$-Ex on the redeposited Pt-Te nanoclusters should be responsible for the production of $CH_x$*, to match the nearly constant probability of conversion to $CH_x$* (Fig. 4b). Earlier studies showed that the C-O bond scission of methanol and subsequent production of $CH_4$ were promoted on supported nanoscale Pt clusters[34]; meanwhile, the dehydrogenation to CO also occurred, which contrasts with the present observation. Therefore, the Pt-Te nanoclusters, instead of pure Pt clusters, are more likely the structures to yield $CH_x$*. These Pt-Te nanoclusters had no long-range structural ordering since our RHEED measurements showed no additional diffraction patterns. As they were formed by nucleation of redeposited Pt and Te, the Pt atoms in the clusters were not bonded exclusively to Te so not separated as far as those in PtTe₂. The $Pt_{uc}$-Ex at the cluster's surface thus differs structurally from the $Pt_{uc}$-Vac and the $Pt_{uc}$-Ex at edges of PtTe₂ patches discussed above. Small Pt aggregates likely formed in the Pt-Te clusters so their surface $Pt_{uc}$-Ex exhibited reactivity partially resembling that of supported Pt nanoclusters[34].

## Discussion

With STM, PES, NAP-PES, NAP-MS, and DFT calculations, we studied methanol reactions on layered PtTe₂ under both UHV and NAP (up to 0.1 mbar) conditions. A structurally perfect PtTe₂ surface was inactive

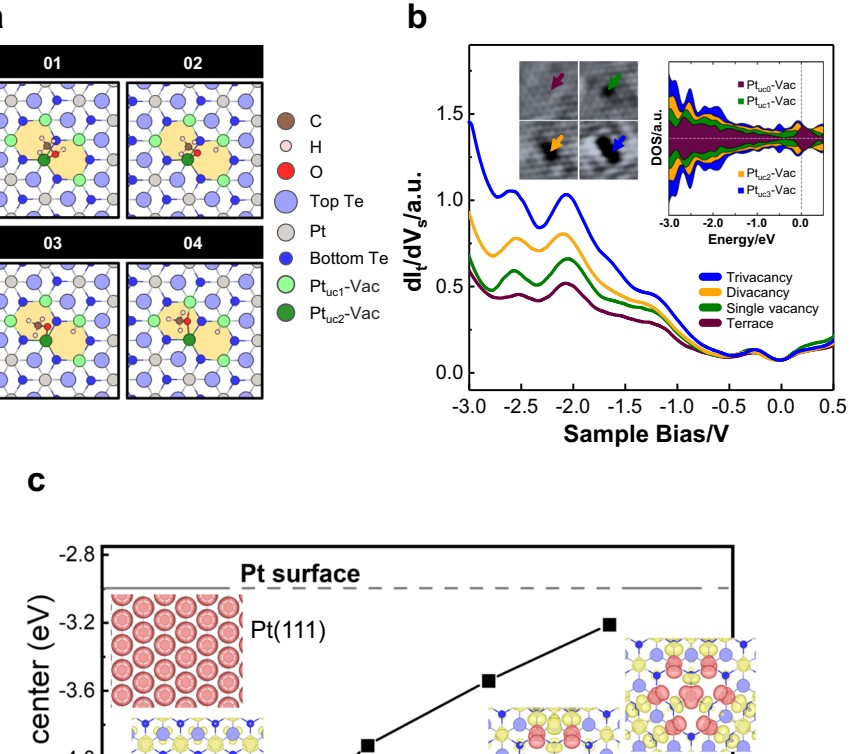

**Fig. 7 | Dehydrogenation processes of CH₃OH\* to CH₃O\* on a Te-divacancy site and electronic structures near the Fermi level of Pt$_{uc}$-Vac. a** Schematics illustrating the dehydrogenation processes (1 → 4) of CH₃OH\* to CH₃O\* on a Te-divacancy site; the corresponding energy profile is given in Fig. S7. **b** Comparison of the measured and calculated (inset) LDOS near the Fermi level of Pt$_{uc0-3}$-Vac. **c** Comparison of d-band centers of Pt$_{uc}$-Vac at PtTe₂ surface and Pt at Pt (111) surface. In **b**, the measurements, dI$_t$/dV$_s$ vs. V$_s$, were conducted with scanning tunneling spectroscopy (STS), the arrows in the STM images (inset) indicate the locations where the STS measurements were performed, and the dash line in the calculated LDOS (inset) indicates the Fermi level. In **c**, the Pt under-coordination number, the x-axis, indicates the number of missing Te-Pt bonds of Pt$_{uc}$-Vac. The insets show the corresponding spatial distributions of frontier orbitals (in the energy ranging from −0.25 eV to the Fermi level) of Pt$_{uc}$-Vac (red spheres), the other atoms (yellow) at PtTe₂ surface, and Pt at Pt (111) surface (red).

whereas the surface with Pt$_{uc}$ sites, which was formed primarily by removing surface Te using controlled Ar⁺ bombardment, became able to activate methanol decomposition. Adsorbed methanol on the Pt$_{uc}$-Vac sites, the dominating reactive sites at small Pt$_{uc}$ concentrations, began to decompose at approximately 160 K and yielded CH$_x$O\* (x = 2 and 3) as the main intermediates; CH$_x$O\* either desorbed as CH₂O$_{(g)}$ or decomposed further, via the transient formation and subsequent C-O bond scission of CH₂OH\*, to produce CH$_x$\* (x = 1 and 2). The reaction probability on the Pt$_{uc}$-Vac sites exceeded 90 % —— approximately 60 % of the methanol decomposed to CH$_x$O\* and 35 % to CH$_x$\* at 180 K, and the reaction ultimately produced gaseous hydrogen, methane, water, and formaldehyde at elevated temperature. We argue that the Pt$_{uc}$-Vac activated the reaction processes like single-atom catalysts and in a coordinative manner; their triangular positioning and varied degrees of oxidation accounted for the observed characteristic reactivity. Increased Ar⁺ dosage (Pt$_{uc}$ concentration increased to 10 – 20 %) generated structurally different Pt$_{uc}$-Ex, associated with the edges of PtTe₂ patches and re-deposited Pt-Te nanoclusters, even though the Te vacancies remained major at the surface; on such Pt$_{uc}$-Ex sites the probability of decomposition to CH$_x$O\* was selectively decreased. A consistent trend was reflected in the gaseous products from the reaction under NAP conditions. The results suggest that the PtTe₂ surface can serve as a superior catalyst toward methanol decomposition, with advantages of great catalytic reactivity and tunable selectivity.

## Methods
### Sample preparation

Single-crystal PtTe₂ was synthesized by the self-flux method. High-purity Pt (99.99%, 0.35 g, 1.79 mmol) foils and a Te ingot (99.9999%, 4.35 g, 34.09 mmol), obtained from Ultimate Materials Technology Co., Ltd, were mixed in a ratio of 1:17 and sealed in a quartz tube under vacuum at ~ 5 × 10⁻⁵ Torr. The tube was heated to 1000 °C in 12 hours, held there for 24 hours, and then slowly cooled to 500 °C at a rate of −3 to −5 °C. The excess Te was subsequently separated by centrifugation. To improve the crystal quality and remove any residual Te, the crystal was sealed in an evacuated quartz tube again and heated at 450 °C for 100 hours. The average yield of PtTe₂ with this method was 90 - 95% (based on Pt). The grown bulk crystal has a diameter near 8.0 mm; it was cleaved in situ before each experiment.

### Characterization
The UHV experiments were conducted in UHV chambers with a base pressure in the regime of 10⁻¹⁰ Torr. Ar⁺ bombardment was performed

with an acceleration energy of 0.5 –2.0 keV and under a pressure of $5 \times 10^{-6}$ Torr; 0.5 keV was chosen for the sample preparation in catalytic studies. The sample was maintained at 300 K for exposure to $CH_3OH$ and methanol-$d_4$ ($CD_3OD$) at selected pressures. The highly pure $CH_3OH$ and $CD_3OD$ (Merck, 99.8%) were additionally purified by repeated freeze–pump–thaw cycles before being introduced into the experimental chambers. The gas exposure in the present work is reported in Langmuir units (1.0 L = $1 \times 10^{-6}$ Torr s). STM images were acquired at sample temperature 77 K (samples cooled from 300 K in an hour) in constant-current mode using an electrochemically etched tungsten tip, with a sample bias voltage ($V_s$) of −150 mV −−500 mV and a tunneling current ($I_t$) of 1.0 − 3.0 nA. The PES and NAP-PES experiments were conducted at the TLS BL09A2 and BL24A beamlines, respectively, at the National Synchrotron Radiation Research Center (NSRRC) in Taiwan[77,78]. For the former, the incident photon beam (with a fixed energy 410 eV) was normal to the surface and photoelectrons were collected (Scienta R3000) at an angle of 58° from the surface normal; for the latter, the beam (with a fixed energy 380 eV) was incident 56° from the surface normal and the analyzer (SPECS NAP 150), equipped with a 4-stages differential pumping system, was placed normal to the surface. The energy resolution was estimated to be near 0.1 eV and the BE was referred to the Au 4 f core level at 84.0 eV of an Au substrate placed beside the investigated PtTe$_2$ sample. All spectra presented here were normalized to the photon flux. The gaseous reaction products were measured with NAP-MS, comprised of a quadrupole mass spectrometer (Hiden HAL201RC) and a doubly-differentially pumping system with a stainless steel tubing inlet terminated with an aperture (a diameter 0.75 mm) 3 mm from the sample. We used a full-range gauge mounted on the reaction chamber to calibrate the ion current at a selected m/z ratio to the partial pressure of the corresponding product. The presented partial pressures of the gaseous products were derived by subtracting the background, which was collected from as-cleaved PtTe$_2$ surface (inert toward methanol decomposition), under the corresponding methanol pressure.

## DFT calculations

First-principles calculations were performed by using Vienna ab initio simulation package (VASP)[79,80], in the framework of DFT with the projector-augmented wave (PAW) method[81] and the Perdew–Burke–Ernzerhof functional[82] type generalized gradient approximation exchange-correlation functional. For the 1T-PtTe$_2$ calculations, the energy cutoff was 330 eV and the energy convergence criterion was $10^{-4}$ eV. A $12 \times 12 \times 1$ Γ-centered k-point sampling was performed for the Brillouin zone integration until the relative energies converged to a few meV. The optimized lattice constant was 3.91 Å, in good agreement with a previous report of 3.9 Å[83]. We included weak van der Waals (vdW) interaction between adjacent two bilayers using the recently developed SCAN+rVV10[84] correction method that yielded excellent geometric and energetic results at a reasonable computational cost. The 1T-PtTe$_2$ layered structure was then modeled by a $6 \times 6$ supercell with three bilayers of 1T-PtTe$_2$. The bottom bilayer was fixed and the remaining two bilayers were allowed to relax. The adsorption energy ($E_{ads}$) for each adsorbate was evaluated by the following standard formula: $E_{ads} = E_{A+S} - E_S - E_A$, where $E_{A+S}$, $E_S$, and $E_A$ denote the total energies of the adsorbed system, the clean PtTe$_2$ layered system, and the chemical potentials of the corresponding adsorbed species in a gas phase, respectively. The energy barriers were determined using the climbing image nudged elastic band (CINEB) method[85].

## Data availability

Data that support the findings of this study are presented in the main article and Supplementary Information files. Source data are provided with this paper.

## Code availability

DFT simulations were performed by using The Vienna Ab initio Simulation Package (VASP)[79,80].

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

## Acknowledgements

The National Science and Technology Council in Taiwan provided support with Grants No. NSTC 109-2112-M-018-008-MY3 (P.-H. C. and C.-P. C.), MOST 109-2112-M-008-030-MY2 (J.-W. H., L.-H. K. and M.-F. L.), NSTC 110-2112-M-213-010 (Y.-L. L. and Y.-J. H.), NSTC 110-2112-M-A49-013-MY3 (W.-H. Chen, C.-Y. C. and C.-L. L.), NSTC 110-2112-M-A49-022-MY2 (W.-H. Chen, C.-Y. C. and C.-L. L.), NSTC 111-2634-F-A49-008 (W.-H. Chen, C.-Y. C. and C.-L. L.), NSTC 112-2112-M-213-017 (C.-H. W.) and NSTC 112-2124-M-006-009 (C.-N. K. and C.-S. L.) for this work. This work is also partially supported by the "Center for the Semiconductor Technology Research" from The Featured Areas Research Center Program within the framework of the Higher Education Sprout Project by the Ministry of Education in Taiwan. CPU time at Taiwan's National Center for High-performance Computing (NCHC) is greatly appreciated. We also thank Drs. Hung-Wei Shiu and An-Cheng Yang for their technical support.

## Author contributions

M.-F. L., C.-L. L., and J.-P. C. conceived and supervised the project. J.-W. H. and L.-H. K. performed measurements of PES, NAP-PES, NAP-MS, and RHEED, and analyzed the data. Y.-L. L., B.-H. L., C.-H. W. and Y.-J. H. assisted the synchrotron-based experiments. C.-N. K. and C.-S. L. prepared the TMDs samples. W.-H. C. and Q.-Y. Z. performed STM characterization. P.-H. C. and J.-P. C. performed DFT calculations. The text was initially composed by J.-W. H. and M.-F. L., with input from all co-authors. All authors further contributed to the discussion of the experimental work and the final version of the manuscript.

## Competing interests

The authors declare no competing interests.
