## [Peer Review File · Nature Communications]

Investigating the Role of Undercoordinated Pt Sites at the Surface of Layered PtTe₂ for Methanol DecompositionReviewers' comments:

Reviewer #1 (Remarks to the Author):

This manuscript carefully investigates Pt_{1-x}Cu_x concentration-dependent methanol decomposition selectivity in single crystal PtTe₂, and shows the promise of PtTe₂ material for methanol decomposition. The manuscript demonstrated new discoveries in developing catalytic material and revealing catalytic properties. Besides, from the mechanistic viewpoint, DFT simulations provide detailed understanding in the origin of dependence on Pt_{1-x}Cu_x in methanol decomposition. However, (1) the key points in the paper are not clear; (2) some important fundamental concepts are missing in terms of the discussion of Pt_{1-x}Cu_x concentration-dependent methanol decomposition selectivity. Therefore, the major concerns should be addressed or discussed before the manuscript can be further considered. Please see the comments in detail below:

1. The key point (Pt_{1-x}Cu_x concentration-dependent methanol decomposition selectivity) in the manuscript is not clear and convincing. Firstly, it is hard for readers to understand under-coordinated Pt (Pt_{1-x}Cu_x). The structural and electronic information on Pt_{1-x}Cu_x is vague from experimental perspective. For example, STM should provide direct evidence to support the theoretical analysis of the structure of Pt_{1-x}Cu_x active center in PtTe₂ (triangularly positioned Pt_{1-x}Cu_x). Secondly, multiple Ar⁺ dosage or increased Ar⁺ kinetic energy generates more Te vacancies on single crystal PtTe₂ surface, which is also defined as surface Pt_{1-x}Cu_x by authors. However, the author need to confirm if more Ar⁺ dosage or increased Ar⁺ kinetic energy generates the same/similar defects by STM. This is critical to correlate the catalytic selectivity to structure.
2. On page 3, line 54, in the background part, when introducing MoS₂, the advances in the reactivity manipulated by sulfur vacancy should be demonstrated in more details (eg. what, how).
3. On page 9, line 172, despite the literature that reports carbon fragments at particular binding energy on different materials, is there any experiment/theoretical evidence to support these assignments on PtTe₂?
4. On page 10, line 179, will integrated area proportion in y axis of Figure 3e and Figure 4 be better, instead of integrated intensities?
5. On page 12, line2 215-227, more in-depth discussion on dehydrogenation or C-O bond scission in methanol over Pt_{1-x}Cu_x of PtTe₂ should be demonstrated. In Figure 4b, CH_xO and CH_x formation in methanol decomposition is dependent and independent on Pt_{1-x}Cu_x concentration, respectively, which is similar to structure sensitivity and insensitivity. A literature (10.1038/ncomms13057) given here would be useful. Considering this fundamental catalysis concept, it is too fast to come to the current conclusion in the manuscript. More careful geometric and electronic structure analyses of Pt in higher Pt_{1-x}Cu_x concentration (especially >10%) are needed.
6. On page 14, lines 254-255, from my understanding, the speculation should be relating to the analysis of the peak at ~ 288 eV, which is a unique feature when the pressure is 10⁻¹ mbar.

Reviewer #2 (Remarks to the Author):

Catalysing methanol decomposition on ion bombarded PtTe₂ surfaces has been investigated, and found to be effective. Under-coordinated Pt sites are proposed as being responsible for activity and selectivity. My major concern is that contrary to the authors' claim, the catalytically active PtTe₂ surfaces are not well-defined crystalline surfaces, with a given concentration of Te vacancies, but rather disordered, full of discontinuities and ill-defined edges. As a consequence, the detailed modelling and reasoning, based on highly crystalline surfaces comprising Te divacancies is also questionable.

STM measurements show that 0.5 keV irradiation for 30s creates only individual defects (defect clusters). However, already a 3 min. irradiation, at a similar energy, induces a rather disordered surface, with crystalline order limited to a few nanometer range. PES spectra in Fig. 3. are given for 9 min. irradiation, which clearly cannot be interpreted within the model based on the structural data provided by STM with 30s irradiation.

Regarding STM measurements, based on the height profile of defects alone, it is not possible to directly conclude that only one (or few) Te atoms are missing. Changing the imaging bias can easily change the apparent depth/height of such defects. Electronic structure effects need to be considered.

The authors claim that clusters are made of redeposited atoms. How can they be sure that these are Te clusters, since Pt atoms are much more likely to form stable clusters, but in this case, we are back to the well-known case of supported Pt cluster catalysts, known to be active for methanol decomposition.

What about the Pt/Te ratio evolution with ion bombardment?

In summary, the authors have shown that inducing disorder in PtTe₂, improves its catalytic activity for methanol decomposition, which can be tuned with irradiation (disorder degree). This is plausible, in fact to be expected. With limited direct practical relevance, the main goal of such studies is to understand better the relations between structure and catalytic properties. However, based on the data provided in the manuscript, it is not possible to gain a deeper understanding of the process, or reach some solid conclusions regarding the structure of active sites or the underlying catalytic mechanisms.

Authors' response to reviewers' report on manuscript NCOMMS-23-21592 entitled: “Dependence on Under-Coordinated Pt at the Surface of Layered PtTe₂ in the Decomposition of Methanol”

We do thank the reviewers for the careful reading and instructive comments. It is clear from the reviewers' comments that some modifications to the above manuscript are required. In the following, we have itemised our corrections to the text. The reviewers' original comments are italicised and new texts inserted into the manuscript are underlined. Paragraph numbering includes any paragraphs running over from the previous page.

Reviewer 1

This manuscript carefully investigates Pt_{uc} concentration-dependent methanol decomposition selectivity in single crystal $PtTe_2$, and shows the promise of $PtTe_2$ material for methanol decomposition. The manuscript demonstrated new discoveries in developing catalytic material and revealing catalytic properties. Besides, from the mechanistic viewpoint, DFT simulations provide detailed understanding in the origin of dependence on Pt_{uc} in methanol decomposition. However, (1) the key points in the paper are not clear; (2) some important fundamental concepts are missing in terms of the discussion of Pt_{uc} concentration-dependent methanol decomposition selectivity. Therefore, the major concerns should be addressed or discussed before the manuscript can be further considered. Please see the comments in detail below:

1. The key point (Pt_{uc} concentration-dependent methanol decomposition selectivity) in the manuscript is not clear and convincing. Firstly, it is hard for readers to understand under-coordinated Pt (Pt_{uc}). The structural and electronic information on Pt_{uc} is vague from experimental perspective. For example, STM should provide direct evidence to support the theoretical analysis of the structure of Pt_{uc} active center in $PtTe_2$ (triangularly positioned Pt_{uc}). Secondly, multiple Ar⁺ dosage or increased Ar⁺ kinetic energy generates more Te vacancies on single crystal $PtTe_2$ surface, which is also defined as surface Pt_{uc} by authors. However, the author need to confirm if more Ar⁺ dosage or increased Ar⁺ kinetic energy generates the same/similar defects by STM. This is critical to correlate the catalytic selectivity to structure.

Reply: We thank the reviewer for raising the critical issue. We agree that the structural characterization of Pt_{uc} sites is critically important to the present work, but direct STM imaging of Pt_{uc} atoms is not the only approach. Our PES spectra already identified the existence of Pt_{uc} atoms, indicated by a smaller binding energy, 72.5 eV,

for Pt_{4f_{7/2}}, typically assigned to Pt species with unsaturated bonding. At small Pt_{uc} concentrations, the structure of Pt_{uc} site is described with the Te-vacancy model (Figure 6), i.e., the underlying Pt atoms remain almost at the same positions, triangularly positioned, after surface Te are removed. If they were moved, the surface Te atoms around the vacancy and bonded to these Pt_{uc} atoms would move from their original positions. Our STM images (Figure 1g) verify that the positions of these surface Te atoms are little altered. The structural modelling with DFT calculations also confirm the scarcely altered positions of Pt_{uc} atoms, as seen in our model (Figure 6). Moreover, the simulated STM image of the Te vacancies, based on the vacancy model and with DFT calculations, matches well the STM image (Figure 1h). These evidences strongly support our Te-vacancy model. The imaging of Pt_{uc} is not straightforward. Simply tuning bias voltage did not yield the Pt_{uc} image; the imaging of such under-coordinated metal atoms at vacancies at the surface of TMD materials is not found in the current literatures.

We should clarify that the varied Pt_{uc} concentrations for our reaction experiments were prepared only with Ar⁺ kinetic energy of 0.5 keV; no other kinetic energies were used (page 6, paragraph 2, lines 11-13; page 9, paragraph 1, lines 7-10).

At greater Pt_{uc} concentrations, produced by greater Ar⁺ dosages, the Te vacancies increased and concomitantly, structurally different Pt_{uc} emerged and accounted for a fraction of total Pt_{uc}. Such Pt_{uc} (denoted as Pt_{uc}(II)) are in structurally ill-defined areas, as exemplified in Figure 1c; they could be associated with the edges of PtTe₂ patches (islands) or redeposited Pt-Te mixtures. With increase of the Pt_{uc}(II) (increased surface roughness), the reactivity per Pt_{uc} site decreased because the Pt_{uc}(II) was less active. We addressed the dependence on the Pt_{uc} concentration to give a more comprehensive picture, whereas the Pt_{uc}(II) was neither the focus nor the key interest in the present work. Nevertheless, we have added more discussion on possible structures of the Pt_{uc}(II) in the revised manuscript.

- New texts inserted (page 6, paragraph 2, lines 3-6):

“The as-cleaved surface was generally very flat and had few defects, as illustrated in Figure 1a; the hexagonally arranged white spots were topmost Te atoms imaged, and several types of intrinsic defects observed previously with STM,⁵⁵ such as Te and Pt vacancies, were also observed...”

- We have altered the texts (page 7, paragraph 1, lines 3-13) to read:

“Figures 1e-h exemplify a high-resolution image for two single-Te vacancies and the corresponding line profile across one of the vacancies. The vacancy depth about 0.1”

nm (Figure 1f), similar to that of an intrinsic Te vacancy at the surface (Figure S1), and the well overlap of the top Te atoms and vacancies in the model with the imaged ones (Figure 1g) suggest that the vacancies were formed by the removal of the topmost Te atoms, i.e., the surface Te vacancies. As the positions of the surface Te atoms neighboring the vacancies remained nearly unchanged (Figure 1g), the generation of these Te vacancies altered little the positions of underlying Pt atoms. The well match of the STM image with the DFT-simulated one (Figure 1h), produced based on the vacancy model in Figure 1g, corroborates the vacancy structure. Our DFT modelling for the reactions discussed below also adds weight to the argument.”

- We have changed Figures 1g and 1h and revised the caption (Figure 1) accordingly:

“...; (e) the high-resolution image of two single-Te vacancies, (f) the line profile across a single-Te vacancy, (g) the overlap of the two single-Te vacancies model with the imaged ones and (h) the match of the STM image with the DFT-simulated one produced based on the vacancy model in (g). In (g), pink, brown and grey balls denote top, bottom Te and Pt atoms in the topmost PtTe₂ bilayer; the dash-line circles denote single-Te vacancies. In (h), green balls denote the simulated images for the top Te atoms; the simulated image was derived with bias -0.1 V while the images obtained with -0.5 ~ -0.1 V were similar.”

- We have altered the texts (page 12, paragraph 1, lines 11-16) to read:

“At greater Pt_{uc} concentrations (> 10 %) the Pt_{uc} structurally different from those at the Te vacancies had grown and accounted for a fraction of total Pt_{uc}, although they were not resolved in the Pt 4f spectra. Such Pt_{uc} (denoted as Pt_{uc}(II)) sites possessed different catalytic properties so initiated a separate reaction pathway, for which the C-O bond scission was more facilitated so CH_xO* became instable and decreased.”

- We have altered the texts (page 13, paragraph 1, lines 6-18) to read:

“The observation does not simply suggest that the formation of CH_xO*

(dehydrogenation) was structure-sensitive whereas that of CH_x^* (C-O bond scission) was not,⁶⁹⁻⁷¹ because the productions of these two intermediates were not necessarily separate — CH_x^* could be produced largely via the C-O bond scission of CH_xO^* . Besides, both the dehydrogenation and C-O bond scission were considered sensitive to structures.⁶⁹⁻⁷¹ The present result is likely due to enhanced C-O bond scission, which decreases CH_xO^* but increases CH_x^* , and obstructed dehydrogenation, which decreases both CH_xO^* and CH_x^* , on the $\text{Pt}_{\text{uc}}(\text{II})$ sites mentioned above. Our STM measurements show that at a greater Ar^+ dosage (Pt_{uc} concentration $> 10\%$), not only the surface Te vacancies but also structurally ill-defined areas, such as the edges of PtTe_2 patches (islands) or redeposited Pt-Te mixtures (as that shown in Figure 1c), grew. We thus associated the $\text{Pt}_{\text{uc}}(\text{II})$ with the Pt in the structurally ill-defined areas.”

• We have altered the texts (page 17, paragraph 3, lines 2-7) to read:

“We established a Te divacancy model by removing two adjacent Te atoms at the topmost layer in order to mimic a PtTe_2 surface at which the reactive sites consist primarily of the Te vacancies (Figures 1b,e). The model is applicable to the cases of Pt_{uc} concentration $\leq 20\%$ because the Te vacancies remained as the main surface defects despite the growth of the $\text{Pt}_{\text{uc}}(\text{II})$ at a Pt_{uc} concentration $> 10\%$.”

• New texts inserted (page 18, paragraph 1, lines 2-4):

“The structural modelling confirms that the underlying Pt atoms remain nearly at the same positions after the removal of surface Te (Figure 6).”

• We have altered the texts (page 23, paragraph 2, lines 5-6 and page 24, paragraph 1, lines 1-14) to read:

“Adsorbed methanol on the Pt_{uc} sites at surface Te vacancies, the dominating reactive sites at small Pt_{uc} concentrations, began to decompose at approximately 160 K and yielded CH_xO^* ($x = 2$ and 3) as the main intermediates; CH_xO^* either desorbed as $\text{CH}_2\text{O}_{(\text{g})}$ or decomposed further, via the transient formation and subsequent C-O bond scission of CH_2OH^* , to produce CH_x^* ($x = 1$ and 2). The reaction probability on the Pt_{uc} sites exceeded 90% — approximately 60% of the methanol decomposed to CH_xO^* and 35% to CH_x^* at 180 K, and the reaction ultimately produced gaseous hydrogen, methane, water and formaldehyde at elevated temperature. We argue that the Pt_{uc} at the surface Te vacancies activated the reaction processes like single-atom catalysts and in a coordinative manner; their triangular positioning and varied degrees of oxidation accounted for the observed characteristic reactivity. With increased Ar^+ dosage (Pt_{uc} concentration increased to $10 - 20\%$), structurally different $\text{Pt}_{\text{uc}}(\text{II})$ emerged, associated likely with the edges of PtTe_2 patches and re-deposited Pt-Te

mixtures, even though the Te vacancies remained major at the surface; on such Pt_{uc}(II) sites the probability of decomposition to CH_xO* was selectively decreased. A consistent trend was reflected in the gaseous products from the reaction under NAP conditions.”

• We have altered the texts (page 5, paragraph 1, lines 11-16) to read:

“At a small Pt_{uc} concentration (≤ 10 %), methanol on the Pt_{uc} at surface Te vacancies, the dominating surface defects, decomposed at a great probability (> 90 %). With increased Pt_{uc} concentration (10 - 20 %), the probability decreased as the probability of decomposition to CH_xO* was selectively decreased, attributed to the structurally different Pt_{uc} generated with increased Ar⁺ bombardment.”

• We have altered the texts (abstract, lines 6-12) to read:

“A great reaction probability (> 90 %) on the Pt_{uc} sites at surface Te vacancies was observed — approximately 60 % of the methanol decomposed to CH_xO* and 35 % to CH_x*, and attributed to both the triangular positioning and varied degrees of oxidation of the Pt_{uc} at Te vacancies. Prolonged Ar⁺ bombardment (Pt_{uc} concentration > 10 %) generated structurally different Pt_{uc}, associated likely with edges of PtTe₂ patches and re-deposited Pt-Te mixtures, on which the probability of decomposition to CH_xO* was selectively decreased.”

2. *On page 3, line 54, in the background part, when introducing MoS₂, the advances in the reactivity manipulated by sulfur vacancy should be demonstrated in more details (eg. what, how).*

Reply: We have inserted new texts accordingly.

• New texts inserted (page 1, paragraph 1, lines 10-13):

“For instance, the sulfur vacancies at MoS₂ surface, in spite of varied generation approaches (plasma,¹⁰ ion bombardment¹³ or chemical etching¹⁴), facilitated hydrogen evolution reaction, through the mechanism involved with altered surface electronic structures and boosted electric conductivity.^{10,13,14”}

3. *On page 9, line 172, despite the literature that reports carbon fragments at particular binding energy on different materials, is there any experiment/theoretical evidence to support these assignments on PtTe₂?*

Reply: We do not have direct experiment or theoretical evidence for the assignment. Nevertheless, these assignments are typical and extensively accepted in the PES studies, as seen in the cited publications (refs. 58-67). Additionally, the assignments agree well with the observed desorbing species (Figures 5c,d).

4. On page 10, line 179, will integrated area proportion in y axis of Figure 3e and Figure 4 be better, instead of integrated intensities?

Reply: We have plotted the figure as the reviewer suggested (see the attached figure below). We do not think the presentation with proportion better because not all carbon species remained on the surface — a great proportion of them, including adsorbed methanol and reaction products, desorbed. This altered figure could cause confusion and be difficult to explain.

5. On page 12, line2 215-227, more in-depth discussion on dehydrogenation or C-O bond scission in methanol over Pt_{uc} of $PtTe_2$ should be demonstrated. In Figure 4b, CH_xO and CH_x formation in methanol decomposition is dependent and independent on Pt_{uc} concentration, respectively, which is similar to structure sensitivity and insensitivity. A literature (10.1038/ncomms13057) given here would be useful. Considering this fundamental catalysis concept, it is too fast to come to the current conclusion in the manuscript. More careful geometric and electronic structure analyses of Pt in higher Pt_{uc} concentration (especially >10%) are needed.

Reply: The conversion probabilities per Pt_{uc} site of CH_xO^* and CH_x^* (Figure 4b)

indeed show that the former decreased with the Pt_{uc} concentration whereas the latter varied little. Nevertheless, the result does not simply suggest the former was structure-sensitive but the latter was not, because the productions of these two intermediates were not separate but correlated — CH_x^* could be produced largely via the C-O bond scission of CH_xO^* . Besides, both the dehydrogenation (for the formation of CH_xO^*) and C-O bond scission (the formation of CH_x^*) are considered structure-sensitive processes (Refs. 69-71). Therefore, the dehydrogenation could be obstructed while the C-O bond scission be enhanced on the $\text{Pt}_{\text{uc}}(\text{II})$ sites generated at greater Ar^+ dosages, mentioned above, so the conversion probability per site of CH_xO decreased but that of CH_x varied little. We addressed the dependence on the Pt_{uc} concentration to give a more comprehensive picture, whereas the $\text{Pt}_{\text{uc}}(\text{II})$ served as neither the focus nor the key interest in the present work because they were less active. We have inserted new texts to clarify the point raised by the reviewer, inserted new references as suggested by the reviewer and added more discussion on possible structures of the $\text{Pt}_{\text{uc}}(\text{II})$ in the revised manuscript. The DFT modelling for methanol on the edges of PtTe_2 patches has also been included.

- We have altered the texts (page 13, paragraph 1, lines 6-18) to read:

“The observation does not simply suggest that the formation of CH_xO^* (dehydrogenation) was structure-sensitive whereas that of CH_x^* (C-O bond scission) was not,⁶⁹⁻⁷¹ because the productions of these two intermediates were not necessarily separate — CH_x^* could be produced largely via the C-O bond scission of CH_xO^* . Besides, both the dehydrogenation and C-O bond scission were considered sensitive to structures.⁶⁹⁻⁷¹ The present result is likely due to enhanced C-O bond scission, which decreases CH_xO^* but increases CH_x^* , and obstructed dehydrogenation, which decreases both CH_xO^* and CH_x^* , on the $\text{Pt}_{\text{uc}}(\text{II})$ sites mentioned above. Our STM measurements show that at a greater Ar^+ dosage (Pt_{uc} concentration > 10 %), not only the surface Te vacancies but also structurally ill-defined areas, such as the edges of PtTe_2 patches (islands) or redeposited Pt-Te mixtures (as that shown in Figure 1c), grew. We thus associated the $\text{Pt}_{\text{uc}}(\text{II})$ with the Pt in the structurally ill-defined areas.”

- New references inserted:

“69. Liu Z-P, Hu P. General Rules for Predicting Where a Catalytic Reaction Should Occur on Metal Surfaces: A Density Functional Theory Study of C–H and C–O Bond Breaking/Making on Flat, Stepped, and Kinked Metal Surfaces. *Journal of the American Chemical Society* **125**, 1958-1967 (2003).

70. Van Santen RA. Complementary Structure Sensitive and Insensitive Catalytic Relationships. *Accounts of Chemical Research* **42**, 57-66 (2009).

71. van den Berg R, *et al.* Structure sensitivity of Cu and CuZn catalysts relevant to industrial methanol synthesis. *Nature Communications* 7, 13057 (2016).”

- New texts inserted (page 22, paragraph 2, lines 1-10 and page 23, paragraph 1, lines 1-10):

“The above modelling also indicates that enlarging the Te vacancies (increasing the number of more active $\text{Pt}_{\text{uc}3}$) promotes the main reaction processes shown in Figure 6, instead of altering the reaction pathway. It accordingly supports the argument that a separated reaction pathway reflected on either the decreased conversion probability to CH_xO on Pt_{uc} sites (Figure 4b) or the decreased production of $\text{CD}_2\text{O}_{(\text{g})}$ and $\text{D}_{2(\text{g})}$ (Figure 5d) at a greater Pt_{uc} concentration was initiated by the $\text{Pt}_{\text{uc}(\text{II})}$. The $\text{Pt}_{\text{uc}(\text{II})}$ could consist of Pt at the edges of PtTe_2 patches (islands) or in redeposited Pt-Te mixtures. Our DFT modelling for methanol on the edges of PtTe_2 patches (Figures S24-S27) show that CH_3OH^* on such $\text{Pt}_{\text{uc}(\text{II})}$ sites prefers desorption to decomposition, in the light of evidently greater activation energies for the dehydrogenation to CH_3O^* (1.64 eV) and the C-O bond scission (2.07 eV) than that for desorption (0.57 eV). The result implies that increasing the edge $\text{Pt}_{\text{uc}(\text{II})}$ sites decreases the average conversion probability (to both CH_xO^* and CH_x^*) on Pt_{uc} sites, in agreement with the observed behavior of CH_xO^* (Figure 4b). As a result, the $\text{Pt}_{\text{uc}(\text{II})}$ in the redeposited Pt-Te mixtures should be responsible for the production of CH_x^* , to match the nearly constant probability of conversion to CH_x^* (Figure 4b). Earlier studies showed that the C-O bond scission of methanol and subsequent production of CH_4 were promoted on supported nanoscale Pt clusters,³⁴ meanwhile, the dehydrogenation to CO also occurred, which contrasts with the present observation. Therefore, the Pt-Te mixtures, instead of pure Pt clusters, are more likely the structures to yield CH_x^* .”

- New Figure inserted (Figure S24, Supplementary Information):

“Figure S24. Dehydrogenation and dihydroxylation of methanol (CH₃OH) on the edge sites of a PtTe₂ island. The adsorbed methanol first dehydrogenates to methoxy (CH₃O) and then to formaldehyde (CH₂O); the processes have energy barriers of 1.64 and 0.92 eV, respectively. Alternatively, the methanol decomposes via C-O bond scission to yield CH₃ and OH, with an energy barrier of 2.07 eV. The detailed information on the energy barrier calculations is provided in Figures S25 - S27. Also, the adsorption energies of related species are calculated and indicated in the parentheses, with the adsorption configurations shown in the panels. In the light of a much smaller barrier for desorption (0.57 eV), the methanol would prefer desorption to decomposition. Details about this edge-site model can be found in Ref. 2.”

6. On page 14, lines 254-255, from my understanding, the speculation should be relating to the analysis of the peak at ~ 288 eV, which is a unique feature when the pressure is 10⁻¹ mbar.

Reply: The spectral feature centered at 288.5 eV in the NAP-PES C 1s spectra (Figure 5) was assigned to the C 1s signal of gaseous methanol. It was also observed on other samples (*The Journal of Chemical Physics* **158**, 174707 (2023)) and varied in intensity evidently with the methanol pressure. It vanished when the pressure was decreased below 10⁻¹ mbar. Accordingly, we believe the assignment is undisputable.

• New reference inserted:

“73. Liao G-J, *et al.* Decomposition of methanol-d₄ on a thin film of Al₂O₃/NiAl(100) under near-ambient-pressure conditions. *The Journal of Chemical Physics* **158**, 174707 (2023).”

Reviewer 2

Catalysing methanol decomposition on ion bombarded PtTe₂ surfaces has been investigated, and found to be effective. Under-coordinated Pt sites are proposed as being responsible for activity and selectivity.

1. *My major concern is that contrary to the authors' claim, the catalytically active PtTe₂ surfaces are not well-defined crystalline surfaces, with a given concentration of Te vacancies, but rather disordered, full of discontinuities and ill-defined edges. As a consequence, the detailed modelling and reasoning, based on highly crystalline surfaces comprising Te divacancies is also questionable. STM measurements show that 0.5 keV irradiation for 30s creates only individual defects (defect clusters). However, already a 3 min. irradiation, at a similar energy, induces a rather*

disordered surface, with crystalline order limited to a few nanometer range. PES spectra in Fig. 3. are given for 9 min. irradiation, which clearly cannot be interpreted within the model based on the structural data provided by STM with 30s irradiation.

Reply: We thank the reviewer for pointing out the unclear part, whether or not the investigated PtTe₂ surface was crystalline. Our reaction experiments were performed at Pt_{uc} concentrations $\leq 20\%$ (Figure 4), corresponding to the Pt_{uc}/Pt ratio ≤ 0.13 and Ar⁺ dosage ≤ 4.5 (Figure 2c), except extreme cases (very great Ar⁺ dosages), which were studied to explore the trend. On the investigated samples, the surfaces were largely crystalline, indicated by the RHEED measurements (Figure S2), although atomically resolved STM images were not always obtained (examples shown below). The surface treated by 3-mins Ar⁺ bombardment (Figure 1c, mentioned by the reviewer), corresponding about to Ar⁺ dosage 6, is beyond the regime under investigation. At small Pt_{uc} concentrations $\leq 10\%$ (corresponding to the Pt_{uc}/Pt ratio ≤ 0.07 and therefore the Ar⁺ dosage ≤ 2 , sputtering time ≤ 60 s), the PtTe₂ surface is crystalline and has some surface Te vacancies, like those in Figures 1b,e and the attached figure. At Pt_{uc} concentrations 10 - 20 %, although the Te vacancies were still major surface defects, structurally different Pt_{uc} (denoted as Pt_{uc}(II)) emerged. The Pt_{uc}(II) were associated with structurally ill-defined areas, such as the edges of PtTe₂ patches (islands) or redeposited Pt-Te mixtures (Figure 1c). They were less active so served as neither the focus nor the key interest in the present work.

Other STM images for the PtTe₂ surfaces under the reaction investigation, prepared with Ar⁺ dosage < 4.5 (Pt_{uc} concentration < 20 %). (a) the surface has some more surface Te vacancies resembling those in Figure 1b; (b) the surface has some islands edges, in addition to the Te vacancies.

We have emphasized in a few places in the manuscript that the reaction characteristics were from restricted Pt_{uc} concentrations. For instance, the discussion of STM and PES measurements emphasized the control of the structural complexity with appropriate Ar⁺ kinetic energy and dosages (page 6, paragraph 2, lines 11 – 13 and page 7, paragraph 1, lines 1-3; page 9, paragraph 1, lines 7-14); the demonstration of C 1s spectra to reveal methanol decomposition with temperature also employed a small Pt_{uc} concentration (the Pt_{uc}/Pt ratio 0.07, Figure 3); the DFT modelling is applicable to the cases of Pt_{uc} concentration ≤ 20 %, for which the surface Te vacancies remained as the main surface defects (page 17, paragraph 2, lines 4 – 7).

The PES spectra in Figure 2 were collected at a great Pt_{uc} concentration (the Pt_{uc}/Pt ratio near 0.2) to show an evident Pt_{uc} signal, since the spectral features were the same even at smaller Pt_{uc} concentrations for which most reaction experiments were performed. We have replaced Figures 2a,b with ones obtained with a smaller Ar⁺ dosage, and included the spectra obtained with other Ar⁺ dosages (other Pt_{uc} concentrations) as references in Supplementary Information.

We have amended the texts to make these points more clearly.

- We have altered the texts (page 6, paragraph 2, lines 8-13 and page 7, paragraph 1, lines 1-3) to read:

“Continuing to increase either the bombardment time or the Ar⁺ kinetic energy generated more not only the surface Te vacancies but also structural variations, such as island edges and re-deposited atoms (Figures 1c, d). With the aim of correlating structures with reactivity, we chose a kinetic energy of 0.5 keV for incident Ar⁺ and a reduced Ar⁺ dosage (the sample current multiplied by bombardment time) to control structural complexity for reaction experiments — the Te vacancies were produced as the main surface defects while the surface crystallinity (monitored with the RHEED measurements, Figure S2) was largely sustained.”

- New texts inserted (page 8, paragraph 1, lines 13-16):

“As the spectral features for varied Ar⁺ dosages are similar despite varied intensities of the Pt_{uc} features (Figure S3), the Pt_{uc} signals correspond to the Pt_{uc} at surface Te vacancies and also to those at other surface defects generated by greater Ar⁺ dosages.”

- We have altered the texts (page 9, paragraph 1, lines 7-14) to read:

“The Ar⁺ at 0.5 keV was chosen to prepare the sample, as it exhibited the best controllability in producing small Pt_{uc} concentrations (small Pt_{uc}/Pt ratios), warranting the surface Te vacancies as the dominating surface defects for catalytic studies. Our reaction experiments were primarily performed on the PtTe₂ with Pt_{uc}/Pt ratios ≤ 0.13 (Ar⁺ dosage ≤ 4.5, Figure 2c), corresponding to the Pt_{uc} concentration ≤ 20 %; the Ar⁺ dosages (0.5 keV) at 1 and 6 in Figure 2c produced the surfaces resembling those shown in Figure 1b and 1c respectively.”

- Replace Figures 2a,b with those obtained with a smaller Ar⁺ dosage:

“**Figure 2.** PES spectra of (a) Pt 4f and (b) Te 4d core levels from layered PtTe₂ as cleaved and bombarded by Ar⁺ (0.5 keV, 3 mins);...

- New figures inserted (Figure S3, Supplementary Information):

Figure S3. PES spectra of Pt 4f and Te 4d core levels from layered PtTe₂ as bombarded by Ar⁺ (0.5 keV) for (a,b) 9, (c,d) 4 and (e,f) 2 mins. Gray circles denote the spectra and black lines the sum of fitted curves; the signals from intact Pt and under-coordinated Pt (Pt_{uc}) in the layered PtTe₂ are fitted with blue and red lines, respectively. The spectral features for varied Ar⁺ dosages are similar despite varied intensities of the Pt_{uc} signals.”

• New texts inserted (page 9, paragraph 2, lines 4-5):

“Figures 3a,b compare the C 1s spectra for methanol adsorbed on as-cleaved and Ar⁺-bombarded PtTe₂ (Pt_{uc}/Pt ratio = 0.07) at 145 K....”

2. Regarding STM measurements, based on the height profile of defects alone, it is not possible to directly conclude that only one (or few) Te atoms are missing. Changing

the imaging bias can easily change the apparent depth/height of such defects. Electronic structure effects need to be considered.

Reply: We thank the reviewer for reminding us this important issue. We had examined the vacancy defects generated by small Ar⁺ dosages. The defects imaged with either positive or negative biases were like dark holes, as exemplified by the STM images below. These topographic images resemble those of top Te vacancies in literatures (for instance, Ref. 55) and additionally, their apparent depths are almost the same as that indicated in Figure 1f. Therefore, we are sure that most of defects created by small Ar⁺ dosages are top Te vacancies.

• New texts inserted (page 7, paragraph 1, lines 3-4):

“The vacancy depth about 0.1 nm (Figure 1f), similar to that of intrinsic Te vacancy at the surface (Figure S1),...”

3. The authors claim that clusters are made of redeposited atoms. How can they be sure that these are Te clusters, since Pt atoms are much more likely to form stable clusters, but in this case, we are back to the well-known case of supported Pt cluster catalysts, known to be active for methanol decomposition.

Reply: We did not claim that the small clusters, formed by re-deposited atoms, at small Ar⁺ dosages are Te clusters. In our original text, we claimed the re-deposited atoms could be Te or Pt (page 7, lines 137-138). We removed detailed discussion on the re-deposited atoms generated by small Ar⁺ dosages because their quantity was limited and hence their effects on the observed reactions was negligible.

• We have altered the texts (page 7, paragraph 1, lines 15-16) to read:

“As their quantity was limited at small Ar⁺ dosages, for which the reaction experiments were performed, their effects on the reactions were considered negligible.”

4. *What about the Pt/Te ratio evolution with ion bombardment?*

Reply: For most samples under the reaction investigation ($\text{Pt}_{\text{uc}}/\text{Pt}$ ratio ≤ 0.13 , Pt_{uc} concentration $\leq 20\%$, Ar^+ dosage ≤ 4.5), the Pt/Te ratio varied within 5%; the altered percentage is near the errors in the serial PES experiments. The small alteration is expected because the defect concentrations were small.

5. *In summary, the authors have shown that inducing disorder in PtTe_2 , improves its catalytic activity for methanol decomposition, which can be tuned with irradiation (disorder degree). This is plausible, in fact to be expected. With limited direct practical relevance, the main goal of such studies is to understand better the relations between structure and catalytic properties. However, based on the data provided in the manuscript, it is not possible to gain a deeper understanding of the process, or reach some solid conclusions regarding the structure of active sites or the underlying catalytic mechanisms.*

Reply: As we replied above, almost all of our reaction experiments were performed on the PtTe_2 surface predominated by crystalline structures, evidenced by STM and RHEED measurements, except extreme cases (very great Ar^+ dosages), which were studied to explore the trend. The surface Te vacancies played as the major surface defects. Therefore, the structure-reactivity relation can be established.

REVIEWER COMMENTS

Reviewer #1 (Remarks to the Author):

Most of my concerns have been addressed.

However, one more important issue should be carefully considered.

It is still not easy for readers to have a clear picture of the dependence of the under-coordinated Pt on methanol decomposition. For example, the structural and electronic information for the real Pt active site in different Ptuc concentrations is not totally given. The authors claim that at small Ptuc concentrations, the structure of Ptuc site is triangularly positioned, but their valence states are not clearly defined; At greater Ptuc concentrations (>10%), the authors attributed the generated Ptuc to Ptuc(II). I assume this should be Pt²⁺. Besides, the authors claim that such Ptuc(II) are in structurally ill-defined areas. There seems to be a lack of solid evidence to figure this specific structural information out.

Therefore, I think the key points of the manuscript should be further solidified and polished before its acceptance.

Reviewer #2 (Remarks to the Author):

It certainly improved the understanding of the manuscript that the authors discuss all the relevant parameters of the ion irradiation (energy, dosage/current, time).

I do not fully agree with the conclusion that the number of redeposited clusters is so low that their effect on catalytic performance can be safely neglected. However, I find plausible the explanation that small Pt clusters are expected to show different activity than observed in the investigated reactions.

Nevertheless, I would not completely exclude their role in the catalytic process.

Another question the authors should perhaps discuss is how accessible these under-coordinated Pt atoms are to the reactants? I wouldn't exclude the possibility that the Te atoms surrounding the Te vacancy site (and bonding directly to the Ptuc atom) also have a role in the reaction.

Authors' response to reviewers' report on manuscript NCOMMS-23-21592A-Z entitled: “Dependence on Under-Coordinated Pt at the Surface of Layered PtTe₂ in the Decomposition of Methanol”

We do thank the reviewers for the careful reading and instructive comments. It is clear from the reviewers' comments that some modifications to the above manuscript are required. In the following, we have itemised our corrections to the text. The reviewers' original comments are italicised and new texts inserted into the manuscript are underlined. Paragraph numbering includes any paragraphs running over from the previous page.

Reviewer 1

Most of my concerns have been addressed. However, one more important issue should be carefully considered. It is still not easy for readers to have a clear picture of the dependence of the under-coordinated Pt on methanol decomposition. For example, the structural and electronic information for the real Pt active site in different Pt_{uc} concentrations is not totally given. The authors claim that at small Pt_{uc} concentrations, the structure of Pt_{uc} site is triangularly positioned, but their valence states are not clearly defined; at greater Pt_{uc} concentrations (>10%), the authors attributed the generated Pt_{uc} to Pt_{uc(II)}. I assume this should be Pt²⁺. Besides, the authors claim that such Pt_{uc(II)} are in structurally ill-defined areas. There seems to be a lack of solid evidence to figure this specific structural information out.

Reply: The structural information on the active Pt_{uc} sites had been addressed in the texts. We attempt to make it more clear in this response. At a small Pt_{uc} concentration ($\leq 10\%$), the Pt_{uc} at surface Te vacancies are the exclusive active sites; their structures are characterized with the STM (page 7, paragraph 2, lines 1-13) and PES (page 8, paragraph 2, lines 1-16 and page 9, paragraph 1, lines 1-3) measurements and confirmed by the DFT modelling (page 18, paragraph 1, lines 7-13). These Pt_{uc} lost 1 – 3 Pt-Te bonds (denoted as Pt_{uc1-3} in the texts) but remained triangularly positioned, as they were before the removal of surface Te, exemplified in the STM images (Figures 1e-h), divacancy (Figures 6a) and trivacancy models (Figure S20). We show below a new atomic model for the Pt_{uc} at a Te divacancy (Figures 6a), as an example. The modelled reaction processes had also been given in Figures S6 – S18, Figures S21 – S22 and Figures S25 – S27. These Pt_{uc}, rather than a single Pt_{uc}, at the vacancies activate the reaction collaboratively, starting with the adsorption of methanol through its O bonded to the Pt_{uc2(or 3)}. The Pt_{uc} can certainly exist at Te vacancies of varied sizes, but our DFT modelling indicates that preferential

reaction pathways were the same (page 22, paragraph 1, lines 3-12, page 23, paragraph 2, lines 1-3 and Figures S20-S22). We, accordingly, have inserted new figures (Figure 6a and Figure 7a) to illustrate the divacancy model and how these Pt_{uc} at the vacancies activate the reaction.

By using the structures of such active Pt_{uc} , we had also computed and compared the local density of states (LDOS) near Fermi levels of Pt at structurally perfect PtTe_2 and $\text{Pt}_{\text{uc}1-3}$ at Te vacancies (Figure S23), and plotted the shift of *d*-band centers and spatial distributions of frontier orbitals (in the energy ranging from -0.25 eV to the Fermi level) of Pt_{uc} (Figure 7c), to characterize the valence states associated with the reaction and hence rationalize the observed reactivity. The characteristic shift of *d*-band centers was typically used and accepted to explain the catalytic performance (Refs. 74 - 76); the spatial characteristics of these frontier orbitals near the Fermi level play a vital role in adsorption and catalytic activities (Ref. 76). The $\text{Pt}_{\text{uc}1-3}$ also exhibited increased metallic properties with decreased coordination number (Figure S23). The related description had been given in the text (page 22, paragraph 1, lines 1-7 and page 23, paragraph 1, lines 1-10). The experimental characterization of such valence states has been conducted with scanning tunneling spectroscopy (STS). The measured valence states (shown below) suggest enhanced electronic densities near the Fermi level at Pt_{uc} sites and therefore more metallic Pt (Pt_{uc}), so confirm our modelling (Figure 7b). These valence states alone cannot represent a characteristic electronic structure responsible for the reactivity since they are determined by the particular atomic structure of Pt_{uc} at Te vacancies. As described in our text (pages 21 – 23), the observed reactivity results from both structural (triangularly positioning) and electronic (partially oxidation) effects of the Pt_{uc} at Te vacancies. We have inserted a new figure (Figure 7b) for the comparison of measured and computed valence states (LDOS near the Fermi level).

At a greater Pt_{uc} concentration (> 10 %), both the Pt_{uc} at Te vacancies and structurally different Pt_{uc} (denoted as $\text{Pt}_{\text{uc}}(\text{II})$) co-existed at the surface. “(II)” denotes the structurally second kind of Pt_{uc} , instead of Pt^{2+} , which had been described in the text (page 12, paragraph 2, lines 16-18, and page 13, paragraph 1, lines 1-3). The $\text{Pt}_{\text{uc}}(\text{II})$ are associated with the other surface defects produced only at greater Ar^+ dosages, including edges of PtTe_2 patches (islands) and Pt-Te nanoclusters formed by de-deposited Pt and Te, exemplified in Figure 1c and Figure S4b. The $\text{Pt}_{\text{uc}}(\text{II})$ was less active so with increased $\text{Pt}_{\text{uc}}(\text{II})$ (increased surface roughness), the average conversion probability (to both CH_xO^* and CH_x^*) on Pt_{uc} sites decreased (Figure 4b). With the support of DFT modelling, we had explained in the text (page 23, paragraph 2, lines 6-9 and page 24, paragraph 1, lines 1-5, and Figures S24-S27) that the $\text{Pt}_{\text{uc}}(\text{II})$ at the edges of PtTe_2 patches was less active. The $\text{Pt}_{\text{uc}}(\text{II})$ in the redeposited Pt-Te

nanoclusters is structurally ill-defined, since the formation of Pt-Te nanoclusters, through the nucleation of re-deposited Pt and Te, was not under the control; they were the side product of Ar⁺ bombardment so not desired in the present study. These nanoclusters could be structurally disordered as our RHEED measurements did not reflect any other ordering structures. We must claim that the Pt_{uc}(II), particularly those at the Pt-Te nanoclusters, were not the key interest in the present work, because their reactivity was incomparable to that of the Pt_{uc} at surface Te vacancies. The motivation of the present work is to search the structures exhibiting the best catalytic properties so to assist the design or fabrication of ideal catalysts. The Pt-Te nanoclusters were neither desired nor controllable in the experiments. Therefore, there is no point to have further characterization of the structures and composition of the Pt-Te nanoclusters. We have revised Figure 4b to clarify the reactivity evolving with the structures,

- New texts inserted (page 6, paragraph 2, lines 6-10):

“After controlled Ar⁺ bombardment, the surface Te vacancies were evidently increased, together with few small clusters on the surface; the size of the vacancies varied from single- to multiple-Te vacancies (Figure 1b) and the proportion of larger vacancies increased with the bombardment time (Figure S1).”

- New texts inserted (page 7, paragraph 2, lines 6-9):

“As the positions of the surface Te atoms neighbouring the vacancies remained nearly unchanged (Figure 1g) and as they were mainly bonded to the underlying Pt, the generation of these Te vacancies altered little the positions of underlying Pt atoms.”

- New texts inserted (page 7, paragraph 2, lines 11-12):

“Small Ar⁺ dosages removed mainly the surface Te while left the underlying Pt atoms unaltered.”

- New texts inserted (page 18, paragraph 1, lines 7-9):

“Although the Te vacancies of various sizes were observed, the divacancy model suffices to represent the key features of the reactions on the Pt_{uc} at the vacancies.”

- New texts inserted (page 21, paragraph 1, lines 4-7):

“Figure 7a shows the simulated dehydrogenation of CH₃OH* to CH₃O* as an example. CH₃OH* first adsorbs on the Pt_{uc2} through its O and then undergoes the scission of O-H bond. At the final stage, H* is bonded to one of Pt_{uc1} and CH₃O* to both Pt_{uc1} and Pt_{uc2}.”

- New figure inserted (Figure 6a):

- New texts inserted (Captions, Figure 6):

“(a) The atomic model for a Te divacancy at PtTe₂ surface. ...In (b), the sizes of Pt and Te atoms in the model are varied to illustrate their relative positions from top view.”

- New figure inserted (Figure 7a):

- New texts inserted (Captions, Figure 7):

“(a) Schematics illustrating the dehydrogenation processes (1 → 4) of CH₃OH* to CH₃O* on a Te-divacancy site; the corresponding energy profile is given in Figure S7.”

- We have altered the texts (page 22, paragraph 2, lines 1-6) to read:
“The corresponding electronic structures reflect the same trend. The measured and calculated local densities of states (LDOS) near the Fermi level of Pt_{tuc0-3} (Figure 7b) show consistently that with decreased coordination number, the Pt_{tuc0-3} at PtTe₂ surface become more metallic, reflected on the enhanced LDOS near the Fermi level; meanwhile, their *d*-band centers shift toward higher energies (Figure 7c). Both results are indicative of enhanced catalytic reactivity.”^{74, 75}”

- New figure inserted (Figure 7b):

b)

- New texts inserted (Captions, Figure 7):

“(b) Comparison of the measured and calculated (inset) LDOS near the Fermi level of Pt_{tuc0-3}. ...In (b), the measurements, dI_t/dV_s vs. V_s , were conducted with scanning tunneling spectroscopy (STS) and the arrows in the STM images (inset) indicate the locations where the STS measurements were performed, and the dash line in the calculated LDOS (inset) indicates the Fermi level.”

- We have altered the texts (page 14, paragraph 1, lines 1-5) to read:

“...at a greater Ar⁺ dosage (Pt_{tuc} concentration > 10 %), the generated surface defects include not only the Te vacancies (Figure 1b) but also other defects, such as edges of PtTe₂ patches (islands) and Pt-Te nanoclusters formed by nucleation of redeposited Pt and Te (exemplified in Figure 1c and Figure S4b).

- Revised Figure 4b:

- New texts inserted (Captions, Figure 4):

“The insets (STM images) in (b) indicate the structural evolution with the concentration of surface Pt_{uc} (Ar^+ dosages).”

Reviewer 2

1. It certainly improved the understanding of the manuscript that the authors discuss all the relevant parameters of the ion irradiation (energy, dosage/current, time). I do not fully agree with the conclusion that the number of redeposited clusters is so low that their effect on catalytic performance can be safely neglected. However, I find plausible the explanation that small Pt clusters are expected to show different activity than observed in the investigated reactions. Nevertheless, I would not completely exclude their role in the catalytic process.

Reply: We claim the effects of redeposited atoms on the reactions were negligible for the samples prepared with “at small Ar^+ dosages” (page 7, paragraph 2, lines 13 – 15 and page 8, paragraph 1, lines 1-3) because their quantity was incomparable to that of the surface Te vacancies. Their number cannot account for the major features of observed reactions, even though they could induce or assist (as an adsorption site) the reactions. Additionally, the redeposited atoms were not exclusively Pt, since small

Ar⁺ dosages removed mainly surface Te, rather than underlying Pt. Small Pt clusters are also not easy to form on PtTe₂ surface, indicated by our DFT calculations (discussed below), due to a significant lattice mismatch.

A tetrahedral Pt₄ cluster was employed to simulate a small Pt cluster on PtTe₂ surface; six potential adsorption configurations of Pt₄@PtTe₂ were considered, as depicted in the figure below. The configuration (a) is the most stable one, with a formation energy of -2.60 eV per atom. Nevertheless, the formation energy of a freestanding Pt₄ cluster is -2.95 eV per atom, lower than those of the six Pt₄@PtTe₂ configurations. This result arises because the Pt-Pt distance at the bottom of Pt₄@PtTe₂ in the preferential configurations is near 4.0 Å (to match the lattice of PtTe₂ surface), which is evidently larger than the typical Pt-Pt distance, near 2.77 Å. This comparison suggests that the formation of small Pt clusters on PtTe₂ surface is challenging; a stable growth nucleus is not easily formed, as Pt, alternatively, could diffuse to the vacancies. Accordingly, the redeposited atoms are more likely to form Pt-Te nanoclusters. Furthermore, we assessed the adsorption behavior of CH₃OH on Pt₄@PtTe₂ in configuration (a). The adsorption energy of CH₃OH on one of the three Pt atoms in contact with PtTe₂ surface was only -0.48 eV, whereas CH₃OH on the top Pt of the Pt₄ cluster was readily dehydrogenated to CH₃O. Once if the small Pt clusters are formed, they could facilitate the reactions. However, as described above, they are too few to dominate the reaction features.

Adsorption of Pt₄ cluster on six distinct sites at PtTe₂ surface; the corresponding formation energy per atom is also presented.

The effects of Pt-Te nanoclusters produced by greater Ar⁺ dosages (at greater Pt_{uc} concentrations) were not neglected; the Pt_{uc}(II) in the redeposited Pt-Te nanoclusters account for the decreased reactivity and the production of CH_x*. We had discussed

their role in the text (page 23, paragraph 2, lines 3-9 and page 24, paragraph 1, lines 1-11). Nevertheless, the Pt-Te nanoclusters were the side product of Ar⁺ bombardment; they were neither desired nor controllable. Their reactivity was incomparable to that of the Pt_{uc} at Te vacancies, so further careful characterization of their structures and composition is not the prior issue in the present work.

- We have altered the texts (page 7, paragraph 2, lines 13-15 and page 8, paragraph 1, lines 1-3) to read:

“The small clusters in Figure 1b likely correspond to re-deposited atoms (0.15 nm, one-atom high) after Ar⁺ bombardment. As they were much fewer than the surface Te vacancies at small Ar⁺ dosages, for which the reaction experiments were primarily performed, their contribution to the observed reactions were considered limited even though they could initiate or assist (as adsorption sites) reactions.”

2. Another question the authors should perhaps discuss is how accessible these under-coordinated Pt atoms are to the reactants? I wouldn't exclude the possibility that the Te atoms surrounding the Te vacancy site (and bonding directly to the Pt_{uc} atom) also have a role in the reaction.

Reply: As the Pt_{uc} atoms are exposed to adsorbing molecules, methanol adsorbs on these Pt_{uc} sites directly or via diffusion, and then undergo reactions, illustrated in Figures 6a,b and Figure S5. As Te atoms at PtTe₂ surface are mainly bonded to the underlying Pt (Refs. 6, 11, 55), instead of their neighbouring surface Te atoms, the removal of surface Te atoms, to generate Te vacancies, altered little the electronic properties of the Te atoms surrounding the vacancies. The property is reflected in the scarcely altered Td 4d spectra after the removal of Te (Figure 2b). The bonding of Te atoms surrounding the Te vacancies thus remains saturated so they remain inert. Our DFT examination for potential adsorption sites (Figure S5) consistently showed that the Te atoms failed to attract methanol or its decomposition intermediates, because of the significantly weaker adsorption on the Te atoms than that on the Pt_{uc}; methanol and its decomposition fragments were not bonded to these surface Te (Figures S4 – S22) but to the Pt_{uc}. Therefore, we are confident that the Te atoms surrounding the Te vacancies are not heavily involved in the methanol decomposition.

- New texts inserted (page 21, paragraph 2, lines 7-14):

“It is noted that neither CH₃O* nor H* is bonded to Te near the vacancy in the process, since they adsorb weakly on these Te sites, like CH₃OH* and its other decomposition intermediates or fragments (Figure S5). As Te atoms at PtTe₂ surface are mainly bonded to the underlying Pt,^{6,11,55} instead of their neighbouring Te atoms, the removal

of surface Te atoms altered little the electronic properties of Te atoms surrounding the vacancies — the bonding of these surface Te atoms remains saturated. As a result, they were not heavily involved in the methanol decomposition.”

REVIEWER COMMENTS

Reviewer #1 (Remarks to the Author):

The updated version of the manuscript has been significantly improved. My previous concerns have been addressed. I recommend its acceptance after further minor revisions.

Another concern is that the structure difference between Ptuc and Ptuc(II) is not clearly demonstrated. At least it is not intuitive. It will be better to clarify it more. I also suggest defining structurally the second kind of Ptuc as another term rather than Ptuc(II) to avoid general confusion.

Reviewer #2 (Remarks to the Author):

The authors addressed the remaining of my concerns in detail. The explanations they have provided are plausible; therefore, I can recommend the manuscript for publication.

Authors' response to reviewers' report on manuscript NCOMMS-23-21592B entitled: “Dependence on Under-Coordinated Pt at the Surface of Layered PtTe₂ in the Decomposition of Methanol”

We do thank the reviewers for the careful reading and instructive comments. It is clear from the reviewers' comments that some modifications to the above manuscript are required. In the following, we have itemised our corrections to the text. The reviewers' original comments are italicised and new texts inserted into the manuscript are underlined. Paragraph numbering includes any paragraphs running over from the previous page.

Reviewer 1

The updated version of the manuscript has been significantly improved. My previous concerns have been addressed. I recommend its acceptance after further minor revisions.

1. Another concern is that the structure difference between Pt_{uc} and $Pt_{uc(II)}$ is not clearly demonstrated. At least it is not intuitive. It will be better to clarify it more.

Reply: We agree with the reviewer. We have inserted new texts to provide more structural information on Pt_{uc} of the second kind (denoted as Pt_{uc}^\dagger in the revised manuscript). The Pt_{uc}^\dagger had two types: (1) Pt_{uc}^\dagger at the edges of $PtTe_2$ patches (Figures 1c and S4b) and (2) Pt_{uc}^\dagger on redeposited Pt-Te nanoclusters (Figure 1c). The former consisted primarily of Pt_{uc1} , the Pt_{uc} with one missing Te-Pt bond, as illustrated in Figure S24. Our DFT modelling (Figures S24-S27) show that CH_3OH^* on the edge Pt_{uc1}^\dagger sites prefers desorption to decomposition, so increasing the edge Pt_{uc}^\dagger sites decreases the average conversion probability (to both CH_xO^* and CH_x^*) on Pt_{uc} sites, which explains the observed behavior of CH_xO^* (Figure 4b). The latter, type (2) Pt_{uc}^\dagger , should be responsible for the production of CH_x^* , to match the nearly constant probability of conversion to CH_x^* (Figure 4b). The Pt_{uc}^\dagger at the Pt-Te nanocluster's surface differ structurally from the Pt_{uc} at Te vacancies and the Pt_{uc}^\dagger at edges of $PtTe_2$ patches discussed above. As the Pt-Te clusters were formed by nucleation of redeposited Pt and Te, the Pt atoms in the clusters were not bonded exclusively to Te so not as separated as those in $PtTe_2$. Small Pt aggregates likely formed in the Pt-Te clusters so their surface Pt_{uc}^\dagger exhibited reactivity partially resembling that of supported Pt nanoclusters (ref. 34). We have revised Figure S24 to illustrate the Pt_{uc}^\dagger at the edges of $PtTe_2$ patches and inserted new texts to explain the structural differences between Pt_{uc} at the Te vacancies and Pt_{uc}^\dagger .

- We have altered the texts (page 23, paragraph 2, lines 6-9 and page 24, paragraph 1, lines 1-6) to read:

“The $\text{Pt}_{\text{uc}}^\dagger$ could consist of Pt at the edges of PtTe_2 patches (Figures 1c and S4b) or in redeposited Pt-Te nanoclusters (Figure 1c). The $\text{Pt}_{\text{uc}}^\dagger$ at the edges of PtTe_2 patches (islands) are primarily $\text{Pt}_{\text{uc}1}$, the Pt_{uc} with one missing Te-Pt bond, as illustrated in Figure S24. Our DFT modelling for methanol on the edges of PtTe_2 patches (Figures S24-S27) show that CH_3OH^* on the edge $\text{Pt}_{\text{uc}}^\dagger$ sites prefers desorption to decomposition, in the light of evidently greater activation energies for the dehydrogenation to CH_3O^* (1.64 eV) and the C-O bond scission (2.07 eV) than that for desorption (0.57 eV). Notably, the activation energies are greater and the adsorption is weaker than those (Figure 6b and Figure S20) on the $\text{Pt}_{\text{uc}2}$ (or $\text{Pt}_{\text{uc}3}$) in the divacancies (trivacancies).”

- Figure S24 revised (including both side and top views for the model presentation):

- New texts inserted (page 24, paragraph 1, lines 15-19 and page 25, paragraph 1, lines 1-3):

“These Pt-Te nanoclusters had no long-range structural ordering since our RHEED measurements showed no additional diffraction patterns. As they were formed by nucleation of redeposited Pt and Te, the Pt atoms in the clusters were not bonded exclusively to Te so not separated as far as those in PtTe_2 . The $\text{Pt}_{\text{uc}}^\dagger$ at the cluster’s surface thus differ structurally from the Pt_{uc} at Te vacancies and the $\text{Pt}_{\text{uc}}^\dagger$ at edges of PtTe_2 patches discussed above. Small Pt aggregates likely formed in the Pt-Te clusters

so their surface Pt_{uc}^\dagger exhibited reactivity partially resembling that of supported Pt nanoclusters.³⁴”

2. *I also suggest defining structurally the second kind of Pt_{uc} as another term rather than $Pt_{uc}(II)$ to avoid general confusion.*

Reply: We agree with the reviewer. We have altered the notation for the second kind of Pt_{uc} to Pt_{uc}^\dagger accordingly to avoid confusion.

Reviewer 2

The authors addressed the remaining of my concerns in detail. The explanations they have provided are plausible; therefore, I can recommend the manuscript for publication.

Reply: We appreciate the reviewer’s efforts in reviewing our manuscript.